# PREFERENCE CONDITIONED MULTI-OBJECTIVE REINFORCEMENT LEARNING: DECOMPOSED, DIVERSITY-DRIVEN POLICY OPTIMIZATION

## ABSTRACT

Multi-objective reinforcement learning (MORL) aims to optimize policies in environments with multiple, often conflicting objectives. While a single, preference-conditioned policy offers the most flexible and efficient solution, existing methods often struggle to cover the entire spectrum of optimal trade-offs. This is frequently due to two underlying challenges: destructive gradient interference between conflicting objectives and representational mode collapse, where the policy fails to produce diverse behaviors. In this work, we introduce $D^3PO$, a novel algorithm that trains a single preference conditioned policy to directly address these issues. Our framework features a decomposed optimization process to encourage stable credit assignment and a scaled diversity regularizer to explicitly encourage a robust mapping from preferences to policies. Empirical evaluations across six standard MORL benchmarks show that $D^3PO$ discovers more comprehensive and higher-quality Pareto fronts, establishing a new state-of-the-art in terms of hypervolume and expected utility, particularly in complex and many-objective environments.

## 1 INTRODUCTION

Reinforcement learning (RL) has emerged as a powerful framework for training agents to make sequential decisions in complex environments. In the standard single-objective setting (SORL), an agent interacts with an environment to maximize the expected cumulative return of a *single scalar reward function*, which encodes the task's objective (Sutton & Barto, 1998). This paradigm has achieved remarkable success in domains ranging from robotics and game playing to recommendation systems and industrial control.

However, many real-world applications do not have a single objective. Instead, they require agents to simultaneously optimize multiple objectives that may be *synergistic, conflicting, or context-dependent*. For example, an autonomous vehicle must trade off between speed, safety, fuel efficiency, and passenger comfort. A logistics agent may need to balance delivery speed against cost and environmental impact. In such scenarios, optimizing a single reward function collapses the richness of the task, often leading to suboptimal or unsafe behaviors. This motivates the field of *Multi-Objective Reinforcement Learning (MORL)*.

MORL extends the RL paradigm by decomposing all objectives with a *vector of reward signals*, where each element of the vector corresponds to a different objective. It is possible that objectives conflict, such that improving the reward in one objective reduces the reward in another. Thus, a single policy cannot capture a global optimum (all objectives are maximized). Instead of learning a single optimal policy, the goal is to learn a set of Pareto-optimal policies. A policy is Pareto-optimal if no other policy exists that can improve at least one objective without worsening any other objective (Felten et al., 2024). Users can then select policies that align with their preferences, typically through *weight vectors* over the objectives (Rodriguez-Soto et al., 2024). This setup enables *preference-driven decision making* and provides flexibility for downstream deployment (Agarwal et al., 2022).

Yet, MORL introduces *fundamental algorithmic and representational challenges* that go beyond those in SORL. A major difficulty lies in the *non-uniqueness of optimal solutions*: the agent must learn to act optimally under multiple, often contradictory reward structures. This requires reasoning about trade-offs and responding to a potentially infinite set of preference queries (Felten et al., 2024).

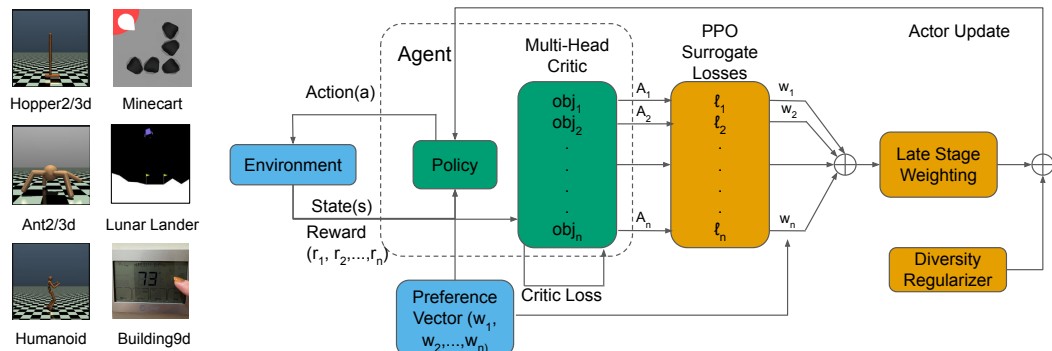

Figure 1: Overview of the $D^3PO$ framework. The architecture decouples credit assignment from preference integration to prevent gradient interference. **(1) Multi-Head Critic:** The critic estimates independent per-objective values $V^{(i)}(s, \omega)$ to compute unweighted advantages $A^{(i)}$. **(2) PPO Surrogate Losses:** The clipping mechanism is applied to each advantage stream *independently* Eq. 2, stabilizing the learning signal before scalarization. **(3) Late-Stage Weighting:** Preference weights $\omega$ are applied only to the stabilized surrogate losses Eq. 4, ensuring gradients are not cancelled prior to optimization. **(4) Diversity Regularizer:** A diversity loss Eq. 3 is added to force behavioral separation between different preference queries, preventing mode collapse.

Furthermore, when objectives conflict, gradients derived from different reward signals may point in opposing directions, *destabilizing policy updates and impairing sample efficiency* (Liu et al., 2025a).

To cope with these challenges, existing MORL approaches have introduced various strategies. However, many contemporary methods face persistent limitations that hinder their performance and scalability. First, methods that learn a single policy often suffer from **destructive gradient interference**: naively combining conflicting objectives into one learning signal produces opposing gradients, so an update that improves one objective can harm another, leading to training instability and suboptimal trade-off policies (Liu et al., 2025a). Second, preference-conditioned policies frequently exhibit **incomplete front coverage** through mode collapse, where the network learns to produce only a small set of similar behaviors for a wide range of preferences, leaving large portions of the Pareto front unexplored. Finally, multi-policy approaches that train a collection of separate policies to cover the front suffer from **architectural inefficiency**, scaling poorly with the number of objectives and incurring significant training and memory costs that make them impractical for complex problems.

We contribute a novel framework, depicted in Figure 1, for training a single, generalizable multi-objective policy that is stable, scalable, and versatile. Our core contributions are:

- **Decomposed Optimization Framework:** We compute unweighted, per-objective advantages and apply preference weights only to the final policy losses. This late-stage weighting decouples preference integration from the core PPO stabilization mechanism, mitigating gradient interference and improving training stability.
- **Scaled Diversity Regularization:** We introduce a loss term that encourages the policy's behavioral divergence, measured via KL divergence, to be proportional to the distance between input preference vectors. This prevents representational mode collapse and promotes the discovery of a diverse Pareto front.
- **A Unified and Scalable Architecture:** The synergy of these components yields a single policy network that generalizes across the entire preference space. Our experiments show this architecture achieves state-of-the-art performance, particularly in complex and many-objective scenarios where prior methods often struggle.

## 2 RELATED WORK

Multi-objective reinforcement learning (MORL) has developed along several algorithmic paradigms, each with distinct strengths and limitations.

**Scalarization.** A foundational approach is scalarization, which reduces vector rewards to a scalar for standard RL methods. Linear scalarization (e.g., weighted sums) is computationally efficient but limited to the convex regions of the Pareto front. Nonlinear scalarization functions (Agarwal et al., 2022; Rodriguez-Soto et al., 2024; Peng et al., 2025) extend expressivity but still collapse objectives into a single training signal, risking loss of information and instability when objectives conflict.

**Multi-policy methods.** Other work trains a set of specialized policies for different preferences, then approximates the Pareto front directly (Cai et al., 2023; Liu et al., 2025c; Hu & Luo, 2024; Yang et al., 2025). Such approaches often rely on constrained optimization or advanced multi-objective optimization techniques to achieve high-quality fronts, but scale poorly with the number of objectives due to the cost of maintaining many policies.

**Decomposition Based Approaches.** Reward- and value-decomposition methods form an influential class of approaches in multi-objective reinforcement learning. These methods explicitly learn objective-specific value functions or successor features and recombine them, typically through generalized policy improvement (GPI), to derive policies for different scalarizations without retraining Barreto et al. (2016; 2019). Variants based on linear scalarization similarly maintain separate per-objective Q-functions and construct policies by applying improvement operators over decomposed value components Van Moffaert & Nowé (2014). More recent work has enhanced GPI-based schemes by prioritizing which decomposed components to update in order to improve sample efficiency Alegre et al. (2023). While such approaches can be effective, they typically rely on linear recombination assumptions and require maintaining multiple value components or policies, and can incur significant storage/compute overhead and limited smooth interpolation across the middle of the Pareto front.

**Single universal policies.** To avoid training multiple policies, recent methods learn a single policy conditioned on a preference vector, enabling adaptation at runtime (Yang et al., 2019; Reymond et al., 2022; Basaklar et al., 2023; Liu et al., 2025a; Kanazawa & Gupta, 2023). Examples include Pareto-Conditioned Networks (PCN) (Reymond et al., 2022), which reuse past transitions across preferences for sample efficiency; Preference-Driven MORL (PD-MORL) (Basaklar et al., 2023), which combines preference conditioning with off-policy engineering such as replay and HER to scale to continuous control; and latent-conditioned policy gradients (Kanazawa & Gupta, 2023), which embed preferences in a latent space. Other PPO-style explorations (e.g., MOPPO (Terekhov & Gulcehre, 2024)) study empirical design choices for conditioned PPO variants. These methods demonstrate the practicality of universal preference-conditioned agents but largely lack formal guarantees against gradient interference or representational collapse.

**Our contribution.** D3PO belongs to this fourth family but differs in two key respects, represented by the orange boxes in Figure 1 : (i) it is an *on-policy* PPO extension with a multi-head critic that preserves raw per-objective signals and applies preferences only after PPO stabilization (Late-Stage Weighting), and (ii) it introduces a *scaled diversity* regularizer that provides formal guarantees against mode collapse. This combination of decomposed advantage preservation, principled preference integration, and provable diversity offers a theoretically enriched alternative to prior preference-conditioned methods, which have primarily emphasized empirical or off-policy approaches.

## 3 PRELIMINARIES

We model decision-making problems with multiple objectives using a *Multi-Objective Markov Decision Process* (MOMDP), formalized as the tuple: $\mathcal{M} = \langle \mathcal{S}, \mathcal{A}, P, R_{1:d}, \Omega, \gamma \rangle$, where $\mathcal{S}$ is the state space, $\mathcal{A}$ is the action space, $P(s' \mid s, a)$ is the transition probability function, $R_i(s, a)$ for $i = 1, \ldots, d$ are $d$ objective-specific reward functions, $\Omega := \{\omega \in \mathbb{R}^d_{\geq 0} \mid \sum_{i=1}^d \omega_i = 1\}$ denotes the space of preference weights, and $\gamma \in [0, 1)$ is the discount factor.

At each timestep $t$, the agent observes state $s_t$, chooses an action $a_t$, and receives a reward vector $r_t = [R_1(s_t, a_t), \ldots, R_d(s_t, a_t)]^\top \in \mathbb{R}^d$. Given a preference vector $\omega \in \Omega$, the overall goal is to find a policy $\pi_w$ that maximizes the expected scalarized return: $\mathbb{E}_\pi \left[ \sum_{t=0}^\infty \gamma^t \cdot \omega^\top r_t \right]$. The unweighted vector return corresponding to a policy $\pi$ is given by: $G^\pi := \mathbb{E}_\pi \left[ \sum_{t=0}^\infty \gamma^t r_t \right]$.

**Pareto Optimality.** Since no single policy can be optimal for all preferences simultaneously, the goal of MORL is to approximate the *Pareto front*—a set of non-dominated policies.

**Definition 1** (Pareto Dominance). *Let $u, v \in \mathbb{R}^d$ be two cumulative return vectors. Then $u$ dominates $v$ (denoted $u \succ v$) if $u_i \geq v_i$ for all $i$, and there exists at least one objective $j$ such that $u_j > v_j$.*

**Definition 2** (Pareto-Optimal Policy). *A policy $\pi$ with a return vector $G^\pi \in \mathbb{R}^d$ is Pareto-optimal if there is no other policy $\pi'$ such that $G^{\pi'}$ dominates $G^\pi$.*

To evaluate MORL algorithms, we use key metrics that quantify both the quality and diversity of the learned Pareto front. **Hypervolume (HV)** measures the volume of the objective space dominated by the discovered front, encouraging both Pareto-dominance and spread. **Sparsity (SP)** measures the evenness of the discovered solutions along the front, with lower values indicating better coverage. **Expected Utility (EU)** measures the average performance across a distribution of sampled preference weights. Together, these metrics assess both the fidelity (HV, EU) and diversity (SP) of the learned solutions.

## 4 METHOD

We propose **Decomposed, Diversity Driven Policy Optimization (D³PO)**, an extension of the standard PPO framework designed to learn a single, unified policy that operates effectively across a continuous spectrum of user-specified preferences. While prior works have explored preference-conditioned policies, they often rely on scalarizing the multi-objective problem prematurely, leading to information loss and challenges with gradient interference between competing objectives. D³PO addresses these limitations by introducing a per-objective optimization framework that maintains the vectorial nature of rewards and advantages throughout the learning process. It promotes the actor to learn different policies for different preferences by introducing a novel diversity driven loss function. This approach enables more stable training and produces a network capable of working with any preference on the simplex $\omega \in \mathbb{R}^d$ s.t. $\sum \omega = 1, \ \omega \geq 0$.

As illustrated in Figure 1, the $D^3PO$ framework operates via a decomposed optimization pipeline designed to prevent gradient interference. The process begins with a **Multi-Head Critic** that estimates independent value functions for each objective, which are used to compute per-objective Generalized Advantage Estimations (GAE). These raw advantage signals are processed individually through **Per-Objective PPO Surrogates** to ensure stability before being aggregated via **Late-Stage Weighting** using the user's preference vector $\omega$. Finally, a **Diversity Regularizer** is added to the actor loss to enforce that distinct preference queries map to distinct behavioral modes.

### 4.1 ARCHITECTURAL AND METHODOLOGICAL INNOVATIONS

The core of D³PO lies in three architectural and methodological innovations that adapt PPO for the multi-objective setting. A detailed summary of the complete method is available in Algorithm 1, found in Appendix A, alongside all Lemmas and Propositions.

**Vectorized Value and Advantage Estimation:** The critic has a multi-head architecture to predict a $d$-dimensional value vector $V(s, \omega) = [V^{(1)}, \ldots, V^{(d)}]$. Consequently, we compute the Generalized Advantage Estimation (GAE) independently for each objective, yielding a $d$-dimensional advantage vector $\mathbf{A}_t$. This preserves the distinct credit assignment signal for each objective. By avoiding premature scalarization, we prevent the *advantage cancellation* formally established in Lemma 1. This can be visualized in Figure 1 as the Multi-head critic with $obj_1, obj_2, \ldots obj_n$ and $A_1, A_2, \ldots, A_n$.

**Decomposed Policy Optimization with Dynamic Sampling:** We compute the PPO clipped surrogate objective for each of the $d$ advantages separately. We then derive the final policy update by multiplying the preference weights and clipped objectives. This ensures that PPO's clipping mechanism operates on the *raw advantage signals*, and the weights $\omega$ are applied only after stabilization. As shown in Proposition 1, this *Late-Stage Weighting (LSW)* preserves the full information content of each advantage stream, and avoids both the destructive cancellation of Early Scalarization (ES) and the premature dampening of Mid-stage Vectorial Scalarization (MVS). This can be visualized in Figure 1 as the PPO Surrogate Losses with $l_1, l_2, \ldots, l_n$, which gets multiplied with $w_1, w_2, \ldots, w_n$ (the objective weights) to construct the final loss.

**Scaled Diversity Regularization:** To prevent mode collapse, we introduce a loss term that increases the policy's behavioral diversity. This works by encouraging the KL divergence between action

distributions to be proportional to the distance between their conditioning preferences. Proposition 3 proves that any minimizer of the resulting actor objective *cannot exhibit representational mode collapse*, ensuring that distinct preferences map to distinct behaviors. This can be visualized in Figure 1 as the Diversity Regularizer, which gets added to the Late Stage Weighting constructed in the prior step.

## 4.2 Per-Objective Advantage and Value Estimation

Following trajectory collection, we compute the GAE for each of the $d$ objective dimensions independently. The critic network, $V_\phi(s, \omega)$, approximates the true state-value vector and is central to this process.

The critic utilizes a multi-head architecture (Figure 1 Green), where a shared network body processes the state $s$ and the preference $\omega$, feeding into $d$ separate output heads. Each head $V_\phi^{(i)}$ is responsible for predicting the **unweighted value** of a single objective $i$. The critic is then updated by minimizing the mean squared error between its predictions and the empirical unweighted returns $G_t^{(i)}$:

$$\mathcal{L}_{\text{critic}}(\phi) = \frac{1}{d} \sum_{i=1}^{d} \mathbb{E}_t \left[ \left( V_\phi^{(i)}(s_t, \omega) - G_t^{(i)} \right)^2 \right]. \tag{1}$$

**Rationale for Conditioning on Preferences.** A key design choice is conditioning the critic $V_\phi(s, \omega)$ on the preference vector $\omega$ even though it predicts unweighted returns. The critic's role is to estimate the state-value function $V_{\pi_\omega}^{(i)}(s)$, which is the expected unweighted return for objective $i$ when following the preference-conditioned policy $\pi(\cdot|s, \omega)$. Since the policy itself is a function of $\omega$, the trajectories it generates and the expected future returns are naturally dependent on $\omega$. Therefore, the critic must be conditioned on $\omega$ to accurately predict these policy-dependent values.

## 4.3 Policy Optimization with Decomposed Gradients and Diversity Regularization

We update the actor network, $\pi_\theta(a|s, \omega)$ (Figure 1 Green), over $K$ epochs for each batch. Our policy optimization combines the standard PPO objective, decomposed per-objective, with a novel diversity-promoting regularizer to enhance the policy's ability to generalize across the preference space.

**Per-Objective Policy Loss:** We first compute the standard PPO clipped surrogate objective independently for each of the $d$ advantage estimates (Figure 1 PPO Surrogate Losses). This isolates the learning signal for each objective before preference application:

$$\mathcal{L}_{\text{clip}}^{(i)}(\theta) = \mathbb{E}_t \left[ \min \left( \rho_t(\theta) A_t^{(i)}, \text{clip}(\rho_t(\theta), 1 - \epsilon, 1 + \epsilon) A_t^{(i)} \right) \right] \tag{2}$$

where the probability ratio is $\rho_t(\theta) = \frac{\pi_\theta(a_t|s_t, \omega)}{\pi_{\theta_{\text{old}}}(a_t|s_t, \omega)}$. As argued in our theoretical analysis, this formulation ensures that PPO's stabilization mechanism is applied to each unweighted advantage, avoiding the signal distortion that plagues ES and MVS.

**Diversity-Promoting Regularization:** Preference-conditioned policies do not always map distinct preference vectors $\omega$ to meaningfully distinct behaviors. To prevent the policy from collapsing to similar strategies for different preferences, we introduce an explicit diversity-promoting loss. During each update, for a given preference $\omega$, we sample a "distractor" preference $\omega'$ by adding small Gaussian noise and re-projecting it onto the preference simplex. (Figure 1 Diversity Regularizer)

We then define a diversity loss that penalizes the policy if the distance between its action distributions, $\pi_\theta(\cdot \mid s_t, \omega)$ and $\pi_\theta(\cdot \mid s_t, \omega')$, does not match the distance between the preferences themselves. We scale the target KL divergence by the L1 distance between the preference vectors:

$$\mathcal{L}_{\text{diversity}}(\theta) = \mathbb{E}_t \left[ \left( D_{KL}(\pi_\theta(\cdot \mid s_t, \omega) \| \pi_\theta(\cdot \mid s_t, \omega')) - \alpha \|\omega - \omega'\|_1 \right)^2 \right]. \tag{3}$$

Proposition 3 shows that minimizing this loss enforces a proportionality between policy divergence and preference divergence, thereby ruling out mode collapse and guaranteeing behavioral diversity.

**Final Actor Objective:** The actor's objective combines two distinct learning signals: (1) a policy improvement term based on the PPO surrogate objective, and (2) our proposed diversity regularizer. To update policy parameters $\theta$, we perform gradient descent on the combined loss function:

$$\mathcal{L}_{\text{actor}}(\theta) = - \left( \sum_{i=1}^{d} \omega_i \mathcal{L}_{\text{clip}}^{(i)}(\theta) \right) + \lambda_{\text{div}} \mathcal{L}_{\text{diversity}}(\theta). \tag{4}$$

Multiplying by the preference weight $\omega_i$ is the critical step translating the user's desired trade-off into a concrete learning signal. Each $\mathcal{L}_{\text{clip}}^{(i)}(\theta)$ represents the raw PPO objective for a single dimension. By scaling each term by its corresponding weight $\omega_i$, we ensure that the final gradient is a weighted sum of the per-objective gradients. This steers the policy update in a direction that prioritizes improving higher weighted objectives, while retaining stability and information preservation guaranteed by Lemma 1 and Proposition 1. The term $\lambda_{\text{div}}$ controls the strength of the diversity regularization, which by Proposition 3 guarantees preference-dependent behavioral separation.

## 5 ANALYSIS OF THE D³PO FRAMEWORK

The success of D³PO arises not from a single algorithmic trick, but from a synergistic framework designed to resolve two fundamental challenges in training a single preference-conditioned policy: (1) achieving **stable credit assignment** in the presence of conflicting objectives, and (2) ensuring the learned policy **generalizes across the preference manifold** rather than collapsing to a limited set of behaviors. Our framework addresses these challenges through three complementary innovations: decomposed value estimation, principled late-stage preference integration, and scaled diversity regularization. Each design choice is motivated by intuition and supported by theoretical analysis, with proofs in the Appendix.

**Stable Credit Assignment via Decomposition:** The first principle of D³PO is *decomposed optimization*, beginning with the critic. The multi-head critic predicts the unweighted expected return $V^{(i)}(s, \omega)$ for each objective $i$, and GAEs are computed independently, yielding a $d$-dimensional advantage vector $\mathbf{A}_t$. This preserves a distinct, interference-free credit signal for each objective.

Intuitively, this avoids contaminating the learning signal with preference-based mixtures too early. Formally, Lemma 1 shows that scalarizing advantages before optimization (as in Early Scalarization, ES) inevitably discards information: the magnitude of the scalarized advantage $|A_t^\omega|$ is strictly smaller than the sum of individual magnitudes whenever objectives conflict. This phenomenon, which we term *advantage cancellation*, explains why ES-based methods (e.g., MOPPO (Terekhov & Gulcehre, 2024)) often stall under conflicting objectives.

**Principled Preference Integration via Late-Stage Weighting:** While decomposition preserves raw signals, preference weighting must still be integrated in a way that avoids distortion. Traditional methods either weight too early (ES) or dampen signals before PPO stabilization (Mid-stage Vectorial Scalarization, MVS). Both approaches risk destructive interference or overly conservative updates.

D³PO instead employs *Late-Stage Weighting (LSW)*: PPO surrogates are computed on raw per-objective advantages, and only the stabilized losses are weighted by preferences. This design decouples PPO's trust region stabilization from user preference scaling: the stabilization mechanism operates on true credit signals, and preferences act only as a final arbitration.

Intuitively, this ensures that PPO "sees" the full significance of each event before preferences adjust its contribution. Formally, Proposition 1 shows that LSW preserves advantage magnitudes while MVS and ES distort them, establishing the robustness hierarchy

$$\text{LSW} \succeq \text{MVS} \succ \text{ES}.$$

This hierarchy guarantees that D³PO avoids gradient interference and remains sensitive to high-magnitude events, even for objectives with low weights. The full proof is in Appendix D. The proof gives a precise mathematical basis for the design choice of LSW: When pipelines include per-objective normalization, per-objective ratios, adaptive clipping, or other non-homogeneous operators (common in practice), LSW preserves stabilized event magnitudes better than MVS (Proposition 2).

**Preventing Collapse via Diversity Regularization:** Stable credit assignment alone is not sufficient. A common failure mode of preference-conditioned agents is *mode collapse*, or "policy laziness,"

where the policy produces nearly identical behaviors across wide regions of the preference simplex. This limits the ability to recover the full Pareto front.

D$^3$PO counters this with a scaled diversity regularizer. During training, a distractor preference $\omega'$ is sampled, and the KL divergence between policies $\pi(\cdot|s,\omega)$ and $\pi(\cdot|s,\omega')$ is penalized if it fails to scale with $\|\omega - \omega'\|_1$. This enforces a structured relationship: small preference changes induce subtle policy shifts, while large changes induce dramatic ones.

Intuitively, this regularizer ensures sensitivity to preferences and prevents collapse to a single *average* policy. Formally, Proposition 3 proves that any minimizer of the combined actor objective cannot exhibit mode collapse: distinct preferences must yield distinguishable action distributions. This is the first formal guarantee of anti-collapse in preference-conditioned MORL.

**Convergence:** Finally, we analyze convergence of the actor updates with LSW and diversity regularization. In the **tabular setting**, Theorem 1 shows that the actor objective is concave in policy probabilities, ensuring global convergence to the optimal policy under exact gradients. In the more **realistic function-approximation setting**, Theorem 2 applies stochastic approximation theory to establish that under standard smoothness, variance, and step-size assumptions, stochastic gradient ascent converges almost surely to stationary points of $J(\theta)$.

This guarantees D$^3$PO is stable in practice and theoretically sound across tabular and neural regimes.

**Synergy and Broader Context:** The strength of D$^3$PO lies in the synergy of these components: *Decomposed value estimation* provides clean, per-objective signals; *Late-Stage Weighting* integrates preferences without interference; *Diversity regularization* ensures generalization and prevents collapse and catastrophic forgetting, which is a problem single-policy techniques suffer.

Together, these components yield a framework that is more robust to advantage cancellation, less prone to collapse, and convergent under standard conditions. Compared to MOPPO, which can suffer from ES's cancellation (Lemma 1), and Pareto-Conditioned Networks, which do not provide collapse guarantees, D$^3$PO introduces a preference-conditioned PPO approach with theoretical support for both stability and diversity.

D$^3$PO trains substantially faster than multi-policy, decomposition-based MORL methods because it learns a single preference-conditioned actor–critic model. All transitions contribute to learning across the full preference space, yielding higher sample efficiency, fewer updates, and lower memory cost than training $K$ separate policies.

D$^3$PO also deploys more reliably. Multi-policy methods require routing or interpolation among a discrete set of trained policies, which becomes brittle when user preferences fall between or outside the trained points. In contrast, D$^3$PO directly maps any continuous preference vector to a valid behavior through $\pi(a \mid s, \omega)$, eliminating routing and ensuring smooth, predictable adaptation across the entire preference space.

## 6 EXPERIMENTS

| Environment | Metrics | PCN | GPI-LS | C-MORL | D$^3$PO |
|---|---|---|---|---|---|
| **Minecart** | HV ($10^2$ ↑) | $5.32 \pm 4.28$ | $6.05 \pm 0.37$ | $6.77 \pm 0.88$ | $\mathbf{7.39 \pm 0.08}$ |
| | EU ($10^{-1}$ ↑) | $1.5 \pm 0.01$ | $\mathbf{2.29 \pm 0.32}$ | $2.12 \pm 0.66$ | $1.9 \pm 0.06$ |
| | SP ($10^{-1}$ ↓) | $0.1 \pm 0.01$ | $0.10 \pm 0.00$ | $0.05 \pm 0.02$ | $\mathbf{0.01 \pm 0.01}$ |
| | CT (↓) | 6 hours | 5 hours | 16 mins | **7 mins** |
| **Lunar Lander-4d** | HV ($10^9$ ↑) | $0.78 \pm 0.17$ | $1.06 \pm 0.16$ | $1.12 \pm 0.03$ | $\mathbf{1.23 \pm 0.04}$ |
| | EU ($10^1$ ↑) | $1.44 \pm 0.37$ | $1.81 \pm 0.34$ | $2.35 \pm 0.18$ | $\mathbf{2.39 \pm 0.19}$ |
| | SP ($10^3$ ↓) | $\mathbf{0.03 \pm 0.23}$ | $0.13 \pm 0.01$ | $1.04 \pm 0.24$ | $0.32 \pm 0.16$ |
| | CT (↓) | 7 hours | 5 hours | 20 mins | **10 mins** |

Table 1: Performance comparison on **discrete** environments (Minecart, Lunar Lander-4d). Metrics: Hypervolume (HV), Expected Utility (EU), Sparsity (SP), and Compute Time (CT).

| Environment | Metrics | CAPQL | PG-MORL | GPI-LS | C-MORL | D³PO |
|---|---|---|---|---|---|---|
| **Hopper-2d** | HV ($10^5 \uparrow$) | $1.15 \pm 0.08$ | $1.20 \pm 0.09$ | $1.19 \pm 0.10$ | $\mathbf{1.37 \pm 0.03}$ | $1.30 \pm 0.03$ |
| | EU ($10^2 \uparrow$) | $2.28 \pm 0.07$ | $2.34 \pm 0.10$ | $2.33 \pm 0.10$ | $\mathbf{2.53 \pm 0.02}$ | $2.47 \pm 0.01$ |
| | SP ($10^2 \downarrow$) | $0.46 \pm 0.10$ | $5.13 \pm 5.81$ | $0.49 \pm 0.37$ | $1.13 \pm 0.19$ | $\mathbf{0.26 \pm 0.31}$ |
| | CT ($\downarrow$) | 3 hours | 8 hours | 12 hours | 36 mins | **20 mins** |
| **Hopper-3d** | HV ($10^7 \uparrow$) | $1.65 \pm 0.45$ | $1.59 \pm 0.45$ | $1.70 \pm 0.29$ | $\mathbf{2.19 \pm 0.32}$ | $2.12 \pm 0.16$ |
| | EU ($10^2 \uparrow$) | $1.53 \pm 0.28$ | $1.47 \pm 0.25$ | $1.62 \pm 0.10$ | $\mathbf{1.81 \pm 0.01}$ | $1.74 \pm 4.9$ |
| | SP ($10^2 \downarrow$) | $2.31 \pm 3.16$ | $0.76 \pm 0.91$ | $0.74 \pm 1.22$ | $0.53 \pm 0.34$ | $\mathbf{0.04 \pm 0.01}$ |
| | CT ($\downarrow$) | 2 hours | 6 hours | 15 hours | 48 mins | **30 mins** |
| **Ant-2d** | HV ($10^5 \uparrow$) | $1.11 \pm 0.69$ | $0.35 \pm 0.08$ | $1.17 \pm 0.25$ | $1.31 \pm 0.16$ | $\mathbf{1.91 \pm 0.18}$ |
| | EU ($10^2 \uparrow$) | $2.16 \pm 0.94$ | $0.81 \pm 0.23$ | $4.28 \pm 0.19$ | $2.50 \pm 0.25$ | $\mathbf{3.14 \pm 0.21}$ |
| | SP ($10^3 \downarrow$) | $\mathbf{0.18 \pm 0.07}$ | $2.20 \pm 3.48$ | $3.61 \pm 2.13$ | $2.65 \pm 1.25$ | $0.66 \pm 0.40$ |
| | CT ($\downarrow$) | 5 hours | 8 hours | 11 hours | 78 mins | **35 mins** |
| **Ant-3d** | HV ($10^7 \uparrow$) | $1.22 \pm 0.33$ | $0.94 \pm 0.12$ | $0.55 \pm 0.81$ | $2.61 \pm 0.26$ | $\mathbf{2.68 \pm 0.21}$ |
| | EU ($10^2 \uparrow$) | $1.30 \pm 0.29$ | $1.07 \pm 0.07$ | $2.41 \pm 0.20$ | $\mathbf{2.06 \pm 0.14}$ | $1.99 \pm 0.08$ |
| | SP ($10^3 \downarrow$) | $0.17 \pm 0.09$ | $0.02 \pm 0.01$ | $1.96 \pm 0.79$ | $0.06 \pm 0.07$ | $\mathbf{0.004 \pm 0.002}$ |
| | CT ($\downarrow$) | 3 hours | 10 hours | 19 hours | 66 mins | **45 mins** |
| **Humanoid-2d** | HV ($10^5 \uparrow$) | $3.30 \pm 0.06$ | $2.62 \pm 0.32$ | $1.98 \pm 0.02$ | $3.43 \pm 0.06$ | $\mathbf{3.76 \pm 0.11}$ |
| | EU ($10^2 \uparrow$) | $4.75 \pm 0.04$ | $4.06 \pm 0.32$ | $3.67 \pm 0.02$ | $4.78 \pm 0.05$ | $\mathbf{5.11 \pm 0.09}$ |
| | SP ($10^4 \downarrow$) | $0^*$ | $0.13 \pm 0.17$ | $0^*$ | $2.21 \pm 3.47$ | $\mathbf{0.003 \pm 0.001}$ |
| | CT ($\downarrow$) | 3 hours | 7 hours | 18 hours | 55 mins | **30 mins** |
| **Building-9d** | HV ($10^{31} \uparrow$) | $4.29 \pm 0.73$ | *T/O* | *T/O* | $7.93 \pm 0.07$ | $\mathbf{8.00 \pm 0.11}$ |
| | EU ($10^3 \uparrow$) | $3.31 \pm 0.06$ | *T/O* | *T/O* | $3.50 \pm 0.00$ | $\mathbf{3.50 \pm 0.003}$ |
| | SP ($10^3 \downarrow$) | $4.34 \pm 3.72$ | *T/O* | *T/O* | $2.79 \pm 0.40$ | $\mathbf{0.03 \pm 0.01}$ |
| | CT ($\downarrow$) | 15 hours | *T/O* | *T/O* | 55 mins | **45 mins** |

Table 2: Performance comparison on **continuous** environments (Hopper, Ant, Humanoid, Building-9d). Metrics: Hypervolume (HV), Expected Utility (EU), Sparsity (SP), and Compute Time (CT). *T/O* indicates timeout after 5 days.

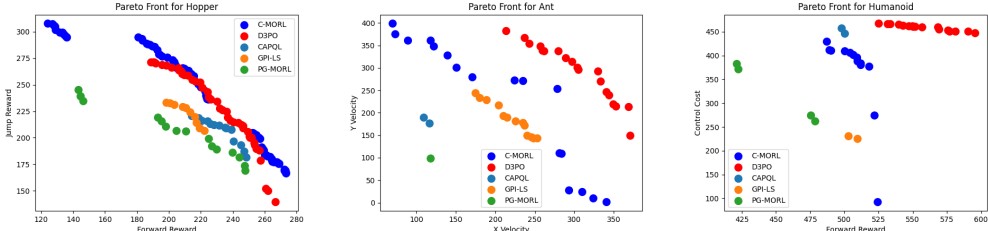

Figure 2: Pareto front comparison on two-objective MO-MuJoCo benchmarks. D³PO (red) discovers a uniform and well-distributed front across the trade-off space, whereas C-MORL (blue) refines extreme points at the cost of higher sparsity. Compared to CAPQL, GPI-LS, and PG-MORL, D³PO achieves broader coverage and reduced collapse, particularly visible in Ant and Humanoid.

We evaluate our proposed method, **D³PO**, against state-of-the-art baselines to answer three key questions: (1) Does D³PO achieve comprehensive Pareto front coverage? (2) Does it effectively prevent mode collapse and generate diverse solutions? (3) Is it computationally efficient?

Our evaluation uses a suite of challenging MORL tasks from the MO-Gymnasium library (Felten et al., 2023), including five continuous control and two discrete control environments, and additionally the Building-9d environment, introduced in Liu et al. (2025b). We compare D³PO against five strong baselines: **PCN** (Reymond et al., 2022), **GPI-LS** (Alegre et al., 2023), **C-MORL** (Liu et al., 2025b), **PG-MORL** (Xu et al., 2020), and **CAPQL** (Lu et al., 2023). For discrete tasks, the number of environment interactions was $5 \times 10^5$ steps. For the more complex continuous control environments, we scaled the number of environment interactions with the number of objectives: $1.5 \times 10^6$, $2 \times 10^6$, and $2.5 \times 10^6$ steps for tasks with two, three, and nine objectives, respectively. We have used the same number of environment interactions as C-MORL (Liu et al., 2025b). We measured performance

with Hypervolume (HV), Expected Utility (EU), Sparsity (SP), and total training Compute Time (CT). Further experimental details are in the appendix.

**D³PO Improves Pareto Front Coverage.** The results in Table 2 and Figure 2 show that D³PO finds dominant and complete solution sets. Quantitatively, D³PO competitively performs (achieves statistically significant improvements) in both Hypervolume and Expected Utility. The significance experiments are analysed in Appendix H.2. In the highly complex MO-Humanoid-2d task, D³PO obtains the highest HV and EU. The advantage is even more pronounced in the nine-objective Building-9d environment, where some baselines (PG-MORL, GPI-LS) failed to complete training within the time limit (5 days). In contrast, D³PO not only finished but also achieved the best HV and EU.

Visually, the Pareto fronts in Figure 2 show D³PO (red) discovering solutions that envelop the baselines. In MO-Ant-2d, for instance, D³PO identifies high-performance "specialist" policies at the extremes of the trade-off space that other methods miss. This superior coverage stems from our core methodological contributions. By computing a vectorized, per-objective advantage and using decomposed policy gradients, D³PO mitigates the destructive gradient interference common in MORL. This process preserves a clean credit assignment signal for each objective, boosting the policy's ability to better exploit the reward landscape and master a wider range of trade-offs.

**Diversity Regularization Prevents Mode Collapse.** A common failure in preference-conditioned MORL is mode collapse, where the policy produces only a single behavior for all preferences. Our second research question investigates how D³PO avoids this.

The most direct evidence is in the MO-Humanoid-2d results (Table 2), where several baselines report a Sparsity (SP) of 0. This indicates a total collapse to a single dominant policy. In contrast, D³PO achieves a low but non-zero SP ($0.003 \times 10^4$), demonstrating that it has learned a diverse and well-distributed set of policies across the front. The visual results in Figure 2 further confirm that D³PO discovers rich, well-spaced pareto fronts.

Diverse policies are primarily due to our proposed scaled diversity regularization. As shown in our ablation study (Table 3), removing the diversity loss (D³PO-DDPO) results in a clear performance drop and, in some cases, collapse to a single-point front (e.g., Humanoid-2d). This highlights that explicitly encouraging the policy to produce distinct behaviors for distinct preferences is critical for discovering a complete and useful Pareto front.

**D³PO Offers Better Computational Efficiency.** Finally, we address the question of efficiency. D³PO is significantly faster than many competing methods because it avoids common computational bottlenecks. Table 2 and 1 shows the total training wall clock time required to train all baselines and D3PO. We can see that D3PO provides a good speedup when compared to the baselines.

Unlike evolutionary or archive-based methods like PG-MORL and CMORL, D³PO does not require an expensive *select-and-improve* loop which selects a solution from a population for further training. Instead, its training process is a continuous, end-to-end optimization analogous to standard PPO, which saves considerable compute time by learning the entire policy manifold simultaneously.

While D³PO consistently achieves competitive results across most benchmarks, we note that C-MORL outperforms on Hopper-2d and Hopper-3d in terms of HV and EU (Table 2). This difference arises from the inherent methodological contrast: C-MORL focuses on iteratively improving existing Pareto solutions, which allows it to refine certain extreme trade-offs and expand the hypervolume. In contrast, D³PO discovers a uniform Pareto front that captures the majority of the trade-off surface but does not fully cover the extremes. As a result, C-MORL attains slightly better HV and EU at the cost of higher sparsity, whereas D³PO maintains lower sparsity and competitive overall coverage. C-MORL's apparent performance differences are not statistically significant(Appendix H.2, indicating that it does not achieve a meaningful advantage over D³PO.

**Qualitative Analysis.** To assess the behavioral diversity captured by $D^3PO$, we developed an interactive interface that allows for real-time modulation of preference weights during rollouts. In the Hopper-3D environment, we observed distinct strategies emerging from extreme weight configurations. A preference of $w = [1, 0, 0]$ (forward velocity) induced rapid forward locomotion, whereas $w = [0, 1, 0]$ (jump height) resulted in high vertical leaps with minimal forward displacement. Conversely, $w = [0, 0, 1]$ (energy efficiency) caused the agent to adopt a stationary stance, minimizing control costs while maintaining upright stability. Furthermore, intermediate weights such as $w =$

| Environment | Metrics | D$^3$PO | D$^3$PO\LSW | D$^3$PO\DDPO |
|---|---|---|---|---|
| Humanoid-2d | HV ($10^5$ ↑) | **3.76 ± 0.11** | 1.50 ± 0.17 | 2.32 ± 0.05 |
| | EU ($10^2$ ↑) | **5.11 ± 0.09** | 2.87 ± 0.22 | 3.83 ± 0.05 |
| | SP ($10^4$ ↓) | **0.003 ± 0.001** | 0$^*$ | 0$^*$ |
| Hopper-2d | HV ($10^5$ ↑) | **1.30 ± 0.03** | 1.23 ± 0.03 | 1.22 ± 0.06 |
| | EU ($10^2$ ↑) | **2.47 ± 0.01** | 2.38 ± 0.05 | 2.42 ± 0.05 |
| | SP ($10^2$ ↓) | 0.26 ± 0.31 | 0.08 ± 0.02 | **0.04 ± 0.02** |
| Ant-2d | HV ($10^5$ ↑) | **1.91 ± 0.18** | 1.53 ± 0.11 | 1.86 ± 0.07 |
| | EU ($10^2$ ↑) | **3.14 ± 0.21** | 2.71 ± 0.13 | 3.09 ± 0.06 |
| | SP ($10^3$ ↓) | 0.66 ± 0.40 | **0.18 ± 0.07** | 0.36 ± 0.09 |

Table 3: Ablation results showing the contributions of Late Stage Weighting (LSW) and Diversity-Driven Policy Optimization (DDPO) in D$^3$PO. LSW improves stability but often collapses the Pareto front (SP = 0), while DDPO preserves diversity and yields more uniform fronts. The full D$^3$PO consistently achieves the best trade-off across HV, EU, and SP.

$[0.5, 0.5, 0]$ produced a seamless interpolation of behaviors, exhibiting velocity and jump height values strictly between those of the extreme policies.

**Ablations.** We introduced two modifications to the actor loss function that allow for the discovery of diverse, evenly spaced Pareto fronts previously inaccessible to single-policy MORL. Thus, we conducted ablation experiments with the Humanoid, Hopper, and Ant environments to understand the impact of our changes. (1) Late Stage Weighting (**LSW**) by multiplying preference weights to the unweighted clipped surrogate objectives to prevent destructive gradient interference. (2) Diversity-driven policy optimization (**DDPO**) by forcing the policy to produce different action distributions scaled by the difference in weights to prevent mode collapse. First, we remove **LSW** by multiplying the preference weights with the advantages after rollout collection, thereby collecting the weighted advantages instead of the unweighted advantages (in effect, ES). In this experiment, we do not remove the diversity loss. Second, we turn off the diversity loss and keep the original decomposed gradient function. In all cases, the critic predicted returns with an expected variance $\approx 1$.

Table 3 shows that both additions are necessary for D$^3$PO's success. Turning off delayed credit assignment (column 2), makes the performance suffer considerably. This shows that learning accurate unweighted returns is necessary to drive correct gradient updates. When we turn on **LSW** and turn off **DDPO** (column 3), we see that the performance improves significantly but it still does not fully approximate the whole front. In both cases, the policies converged prematurely to a single point front in the Humanoid environment. For Hopper and Ant the combination of low HV, EU and SP values for the two cases shows that they discovered an inferior Pareto front compared to D$^3$PO. These experiments show that these additions are necessary to learn robust policies that approximate a high quality Pareto Front. Further, Appendix C presents an ablation over the loss scaling parameter $\lambda_{div}$, showing that while the diversity regularizer itself is essential, the discovered front is robust to the precise value of $\lambda_{div}$. An ablation on the $\alpha$ parameter also shows that the scaling parameter does not affect the results, unless it is explicitly turned off (zeroed out), which results in collapse, or set to a very high value, which diminishes the KL term.

# 7    CONCLUSION

In this work, we introduced D$^3$PO, a novel algorithm for training a single, generalizable policy for MORL. We identified two critical challenges that hinder prior preference-conditioned methods: destructive gradient interference and representational mode collapse. Our proposed framework addresses these issues through a synergy of two principled mechanisms: a decomposed optimization process that preserves the integrity of per-objective credit assignment, and a scaled diversity regularization term that enforces a robust and high-fidelity mapping from the preference space to the policy manifold. Our experiments demonstrate that D$^3$PO performs competitively with the state-of-the-art, discovering more complete and higher-quality Pareto fronts than existing methods, with particularly pronounced advantages in complex, high-dimensional control and many-objective scenarios.

REPRODUCIBILITY STATEMENT

We have taken several steps to ensure the reproducibility of our work. All algorithmic details of $D^3PO$ are fully specified in Section 4, with pseudocode provided in Algorithm 1. Our theoretical results are supported by complete proofs in Appendix D E, where all assumptions are stated explicitly. The experimental setup, including environment details, hyperparameters, and evaluation metrics, is documented in Section 6 and further expanded in Appendix H. We use publicly available benchmark environments without modification, and we describe our training protocols and data processing steps in detail. Anonymous source code implementing $D^3PO$, along with scripts for reproducing all experiments and figures, is included in the supplementary material. Together, these resources ensure that both the theoretical and empirical contributions of this paper are fully reproducible.

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

# Appendix

## A  D³PO PSEUDOCODE

---

**Algorithm 1** Decomposed, Diversity-Driven Policy Optimization

---

**Require:** Actor $\pi_\theta(a \mid s, \omega)$, multi-head critic $V_\phi(s, \omega) \in \mathbb{R}^d$, Optimizers $\mathrm{Opt}_\theta, \mathrm{Opt}_\phi$, and hyper-parameters $\gamma, \lambda, \epsilon, \beta, \lambda_{\mathrm{div}}, \alpha$

1: Initialize network parameters $\theta, \phi$ and rollout buffer $\mathcal{D}$
2: Sample an initial preference vector $\omega$ from the preference space $\Omega$
3: **for** iteration $= 1, 2, \ldots$ **do**
4:     Clear rollout buffer $\mathcal{D}$
5:     **for** $t = 1$ **to** $T$ **do**
6:         Sample action $a_t \sim \pi_\theta(\cdot \mid s_t, \omega)$
7:         Execute $a_t$ and observe next state $s_{t+1}$, reward vector $\mathbf{r}_t \in \mathbb{R}^d$, and done flag $d_t$
8:         Store transition $(s_t, a_t, \mathbf{r}_t, \omega, \log \pi_\theta(a_t \mid s_t, \omega))$ in $\mathcal{D}$
9:         $s_t \leftarrow s_{t+1}$
10:        **if** $d_t$ is True **then**
11:            Reset environment to get new state $s_t$ and resample a new preference vector $\omega \sim \Omega$
12:        **end if**
13:    **end for**
14:    Compute unweighted advantages $\mathbf{A}_t = [A_t^{(1)}, \ldots, A_t^{(d)}]$ and returns $\mathbf{G}_t$ for all transitions in $\mathcal{D}$ using GAE with $V_\phi$.
15:    **for** epoch $= 1$ **to** $E$ **do**
16:        **for** each minibatch $\mathcal{B} \subset \mathcal{D}$ **do**
17:            Let $(s, a, \mathbf{A}, \mathbf{G}, \omega, \log \pi_{\mathrm{old}})$ be the data in $\mathcal{B}$
18:            Predict value vector $\mathbf{V}_\phi(s, \omega) = [V_\phi^{(1)}, \ldots, V_\phi^{(d)}]$
19:            $\mathcal{L}_{\mathrm{critic}} \leftarrow \frac{1}{d} \sum_{i=1}^{d} \left( V_\phi^{(i)}(s, \omega) - G^{(i)} \right)^2$
20:            Update critic parameters $\phi$ using $\mathrm{Opt}_\phi$ and $\nabla_\phi \mathcal{L}_{\mathrm{critic}}$
21:            Sample distractor weights $\omega'$ by perturbing and re-normalizing $\omega$
22:            Compute per-objective PPO losses $\{\mathcal{L}_{\mathrm{clip}}^{(i)}\}_{i=1}^{d}$ using unweighted advantages $\mathbf{A}$
23:            Compute diversity loss $\mathcal{L}_{\mathrm{diversity}}(\theta) = \mathbb{E}_{s \in \mathcal{B}} \Big[ \big( D_{KL}(\pi_\theta(\cdot \mid s, \omega) \| \pi_\theta(\cdot \mid s, \omega')) - \alpha \| \omega - \omega' \|_1 \big)^2 \Big]$
24:            Compute entropy bonus $\mathcal{H} \leftarrow \mathbb{E}_{s \in \mathcal{B}}[\mathrm{H}(\pi_\theta(\cdot \mid s, \omega))]$
25:            $\mathcal{L}_{\mathrm{actor}} \leftarrow - \left( \sum_{i=1}^{d} \omega_i \mathcal{L}_{\mathrm{clip}}^{(i)} \right) - \beta \mathcal{H} + \lambda_{\mathrm{div}} \mathcal{L}_{\mathrm{diversity}}$
26:            Update actor parameters $\theta$ using $\mathrm{Opt}_\theta$ and $\nabla_\theta \mathcal{L}_{\mathrm{actor}}$
27:        **end for**
28:    **end for**
29: **end for**

---

## B  METRICS DEFINITIONS

**Definition 3** (Hypervolume Indicator). *Given a reference point $r \in \mathbb{R}^d$ that all Pareto-optimal returns dominate, the* hypervolume *of a finite set $\{u^k\}$ is, where LM stands for Lebesgue Measure:*

$$\mathrm{HV}(\{u^k\}; r) = LM \left( \bigcup_k \{u \in \mathbb{R}^d : r \le u \le u^k\} \right)$$

**Definition 4** (Sparsity Indicator). *Let $\{u^1, \ldots, u^K\} \subset \mathbb{R}^d$ be an ordered set of Pareto-approximated points. Define the* sparsity *as:*

$$\mathrm{SP}(\{u^k\}) = \frac{1}{K-1} \sum_{k=1}^{K-1} \|u^{(k+1)} - u^{(k)}\|_2$$

**Definition 5** (Expected Utility). *Let $\mathcal{W} \subset \mathbb{R}^d$ be a distribution over preference weights and let $\pi_\omega$ denote the policy conditioned on $\omega$. The* expected utility *is:*

$$\text{EU} = \mathbb{E}_{\omega \sim \mathcal{W}}[\omega^\top G^{\pi_\omega}].$$

**Definition 6** (Compute Time). *The compute time is defined as the time taken by the algorithm to complete its training given the fixed budget of environment interactions. It is calculated as the wall clock time required to complete the entire training pipeline*

## C    EFFECT OF $\lambda_{div}$ AND $\alpha$ ON PARETO FRONT

| Metric | $\lambda_{\text{div}} = 0$ | $\lambda_{\text{div}} = 0.01$ | $\lambda_{\text{div}} = 0.1$ | $\lambda_{\text{div}} = 0.5$ | $\lambda_{\text{div}} = 1.0$ |
|---|---|---|---|---|---|
| HV ($10^5 \uparrow$) | $2.32 \pm 0.05$ | $\mathbf{3.76 \pm 0.11}$ | $3.73 \pm 0.07$ | $3.72 \pm 0.10$ | $3.73 \pm 0.07$ |
| EU ($10^2 \uparrow$) | $3.83 \pm 0.05$ | $\mathbf{5.11 \pm 0.09}$ | $5.08 \pm 0.06$ | $5.07 \pm 0.09$ | $5.07 \pm 0.06$ |
| SP ($10^3 \downarrow$) | $0^*$ | $\mathbf{0.03 \pm 0.01}$ | $0.047 \pm 0.045$ | $0.059 \pm 0.044$ | $0.053 \pm 0.032$ |

Table 4: Ablation results on MO-Humanoid-2d across different values of $\lambda_{\text{div}}$. The results show that the discovered Pareto front remains stable and high-performing over a wide range of $\lambda_{\text{div}}$, indicating robustness of the method to this hyperparameter.

Table 4 reports ablation results on Humanoid-2d across a sweep of $\lambda_{\text{div}}$ values. These results demonstrate that the diversity regularizer itself plays a critical role in shaping the discovered Pareto front. Without diversity encouragement ($\lambda_{\text{div}} = 0$), the algorithm collapses toward limited modes, yielding weaker hypervolume and expected utility despite producing seemingly low sparsity values. Introducing a nonzero regularizer ($\lambda_{\text{div}} > 0$) resolves this issue by preventing mode collapse and maintaining broad front coverage, thereby producing substantially stronger Pareto sets.

At the same time, the quantitative metrics reveal that the performance is relatively insensitive to the precise choice of $\lambda_{\text{div}}$. Across the range $\lambda_{\text{div}} \in \{0.01, 0.1, 0.5, 1.0\}$, hypervolume and expected utility remain consistently high, and sparsity values remain comparable. This indicates that while the presence of the diversity term is essential, its specific scaling does not heavily influence the outcome. Overall, these ablations reinforce that the diversity regularizer is the key mechanism enabling robust front discovery, and that the method is not fragile to the exact tuning of $\lambda_{\text{div}}$.

| Metric | $\alpha = 0$ | $\alpha = 0.1$ | $\alpha = 1$ | $\alpha = 10$ |
|---|---|---|---|---|
| HV ($10^5 \uparrow$) | $2.50 \pm 0.12$ | $3.71 \pm 0.08$ | $\mathbf{3.76 \pm 0.11}$ | $3.20 \pm 0.10$ |
| EU ($10^2 \uparrow$) | $3.90 \pm 0.09$ | $5.03 \pm 0.07$ | $\mathbf{5.11 \pm 0.09}$ | $4.80 \pm 0.27$ |
| SP ($10^3 \downarrow$) | $0^*$ | $0.07 \pm 0.02$ | $\mathbf{0.03 \pm 0.01}$ | $0.12 \pm 0.08$ |

Table 5: Ablation results on MO-Humanoid-2d across different values of $\alpha$.

Table 5 reports similar results. When $\alpha = 0$, the weights scaling parameter is turned off. This keeps the KL term active, and the loss function now tries to minimize the KL. By minimizing the KL, the function actively promotes collapse. Thus, $\alpha$ is an extremely important parameter. When $\alpha = 0.1$ and $\alpha = 1$, the results are similar. This shows that D$^3$PO is robust to the values of the weight parameter. Choosing a very high value $\alpha = 10$ is also detrimental to performance, as that term dominates the loss function. Thus, a reasonable choice for $\alpha$ is between 0.1 and 1.

## D    THEORETICAL ANALYSIS OF MULTI-OBJECTIVE PPO FORMULATIONS

To justify the design of our proposed Late-Stage Weighting (LSW) framework, we provide a formal, unified comparative analysis of three distinct methods for integrating preference weights into the Proximal Policy Optimization (PPO) objective. We prove that LSW is the most robust formulation against the signal distortion caused by conflicting advantages and preference scaling, and we characterize precisely when differences between MVS and LSW arise in practice.

### D.1 Formal Definitions of MORL-PPO Variants

Let

$$\rho_t(\theta) \;=\; \frac{\pi_\theta(a_t \mid s_t, \omega)}{\pi_{\theta_{\text{old}}}(a_t \mid s_t, \omega)}$$

be the importance sampling ratio and $\mathbf{A}_t = [A_t^{(1)}, \ldots, A_t^{(d)}]$ the vector of per-objective advantages. We compare three natural ways to incorporate the preference vector $\omega \in \Delta^{d-1}$ into a PPO-style surrogate.

**Method 1: Early Scalarization (ES).** Scalarize advantages first, then apply the PPO surrogate (Terekhov & Gulcehre, 2024):

$$\mathcal{L}_{\text{clip}}^{ES}(\theta) \;=\; \mathbb{E}_t\Big[\min\big(\rho_t(\theta)\,(\omega^\top \mathbf{A}_t),\; \text{clip}(\rho_t(\theta), 1-\epsilon, 1+\epsilon)\,(\omega^\top \mathbf{A}_t)\big)\Big]. \tag{5}$$

**Method 2: Mid-stage Vectorial Scalarization (MVS).** Form per-objective weighted advantages, apply per-objective surrogates, then sum:

$$\mathcal{L}_{\text{actor}}^{MVS}(\theta) \;=\; -\sum_{i=1}^{d} \mathbb{E}_t\Big[\min\big(\rho_t(\theta)\,(\omega_i A_t^{(i)}),\; \text{clip}(\rho_t(\theta), 1-\epsilon, 1+\epsilon)\,(\omega_i A_t^{(i)})\big)\Big]. \tag{6}$$

**Method 3: Late-Stage Weighting (LSW).** Compute per-objective PPO surrogates on raw advantages and weight the resulting stable surrogate terms:

$$\mathcal{L}_{\text{actor}}^{LSW}(\theta) \;=\; -\sum_{i=1}^{d} \omega_i\, \mathbb{E}_t\Big[\min\big(\rho_t(\theta)\,A_t^{(i)},\; \text{clip}(\rho_t(\theta), 1-\epsilon, 1+\epsilon)\,A_t^{(i)}\big)\Big]. \tag{7}$$

### D.2 Comparative Results

We now formalize the intuition that ES is fragile in the presence of conflicting advantages, show an algebraic equivalence between MVS and LSW under the standard (homogeneous) PPO surrogate, and finally state a provable condition under which LSW is strictly preferable in practical pipelines that include per-objective preprocessing or adaptive, non-homogeneous operations.

**Lemma 1** (ES magnitude loss). *Let $A_t^\omega := \omega^\top \mathbf{A}_t$ and $M_{LSW} := \sum_{i=1}^{d} \omega_i |A_t^{(i)}|$. Then*

$$|A_t^\omega| \leq M_{LSW},$$

*with strict inequality whenever there exist $i, j$ with $A_t^{(i)} A_t^{(j)} < 0$ and $\omega_i, \omega_j > 0$.*

*Proof.* Immediate from the triangle inequality:

$$\big|\omega^\top \mathbf{A}_t\big| = \Big|\sum_{i=1}^{d} \omega_i A_t^{(i)}\Big| \leq \sum_{i=1}^{d} \omega_i |A_t^{(i)}| = M_{LSW}.$$

Strictness follows because the triangle inequality is strict when at least two nonzero terms have opposite signs. $\qquad\square$

**Proposition 1** (Conditional equivalence of MVS and LSW under homogeneous surrogate). *Assume the PPO surrogate evaluates each candidate term by multiplication with a scalar factor drawn from $\{\rho_t(\theta), \text{clip}(\rho_t(\theta), 1-\epsilon, 1+\epsilon)\}$, i.e. the surrogate is homogeneous and linear in the advantage. Under this homogeneity hypothesis, the MVS and LSW actor objectives are algebraically identical:*

$$\mathcal{L}_{actor}^{MVS}(\theta) \;=\; \mathcal{L}_{actor}^{LSW}(\theta).$$

*Proof sketch.* For a fixed objective index $i$ and given scalar multipliers $c_t(\rho) \in \{\rho_t(\theta), \text{clip}(\rho_t(\theta), 1-\epsilon, 1+\epsilon)\}$, the per-objective MVS surrogate is

$$\min\big(c_t(\rho)\,\omega_i A_t^{(i)},\; c_t'(\rho)\,\omega_i A_t^{(i)}\big).$$

Because $\omega_i \geq 0$, the scalar $\omega_i$ factors out:

$$\min\big(c_t(\rho)\,\omega_i A_t^{(i)},\; c_t'(\rho)\,\omega_i A_t^{(i)}\big) = \omega_i \min\big(c_t(\rho)A_t^{(i)},\; c_t'(\rho)A_t^{(i)}\big).$$

Summing over $i$ yields $\mathcal{L}_{\text{actor}}^{MVS}(\theta) = \mathcal{L}_{\text{actor}}^{LSW}(\theta)$, proving algebraic equivalence. $\qquad\square$

**Remark 1.** *At first glance, MVS and LSW appear algebraically similar. Indeed, under the highly restrictive assumption of a homogeneous surrogate with no per-objective preprocessing, they are equivalent. However, this assumption never holds in practice: variance normalization, per-objective critics, and clipping introduce non-homogeneities that make the order of operations critical. In these realistic settings, LSW uniquely preserves the full magnitude of the stabilized advantage signal, while MVS prematurely dampens it.*

**Proposition 2** (Practical superiority of LSW under non-homogeneous per-objective processing)**.** *Suppose some per-objective preprocessing operators $\mathcal{P}_i(\cdot)$ are applied to advantages before the surrogate, where $\mathcal{P}_i$ is not positively homogeneous of degree 1 (i.e., $\exists r_i \neq 1$ such that $\mathcal{P}_i(\alpha x) = \alpha^{r_i} \mathcal{P}_i(x)$ does not hold for all $\alpha > 0$). Then there exist advantages $\{A_t^{(i)}\}$ and weights $\{\omega_i\}$ for which*

$$\omega_i \mathcal{P}_i(A_t^{(i)}) \;\neq\; \mathcal{P}_i(\omega_i A_t^{(i)}),$$

*and, in these cases, weighting* after *stabilization (LSW) preserves a strictly larger stabilized contribution than weighting* before *stabilization (MVS).*

*Proof sketch.* If $\mathcal{P}_i$ is linear and homogeneous of degree 1, then $\mathcal{P}_i(\omega_i A) = \omega_i \mathcal{P}_i(A)$ and no difference arises (cf. Proposition 1). For any $\mathcal{P}_i$ that is nonlinear or homogeneous of degree $r_i \neq 1$, the order of scaling matters. For example, take $\mathcal{P}_i(x) = |x|^\gamma \operatorname{sign}(x)$ (a toy nonlinearity with degree $\gamma$). Then

$$\mathcal{P}_i(\omega_i A) = \omega_i^\gamma |A|^\gamma \operatorname{sign}(A), \qquad \omega_i \mathcal{P}_i(A) = \omega_i |A|^\gamma \operatorname{sign}(A).$$

If $0 < \omega_i < 1$ and $\gamma < 1$, then $\omega_i^\gamma > \omega_i$, so $|\mathcal{P}_i(\omega_i A)| > |\omega_i \mathcal{P}_i(A)|$. Thus there exist realistic preprocessing operators for which applying $\omega_i$ before preprocessing reduces the stabilized magnitude compared to applying $\omega_i$ after preprocessing. Many practical pipelines include variance normalization, adaptive per-objective clipping, or critic-dependent scaling, all of which break degree-1 homogeneity; in these common cases LSW preserves larger stabilized signals than MVS. $\square$

**Corollary 1** (Hierarchy of robustness)**.** *Combining Lemma 1, Proposition 1, and Proposition 2 yields the claimed robustness ordering:*

$$LSW \succeq MVS \succ ES,$$

*where '$\succeq$' denotes practical superiority (LSW is at least as robust as MVS in the homogeneous surrogate and strictly more robust when non-homogeneous per-objective processing is present), and '$\succ$' indicates strict superiority over ES due to avoidance of inter-objective advantage cancellation.*

### D.3 IMPLICATIONS

The above results give a precise mathematical basis for the design choice of LSW:

- **Avoid cancellation:** ES can drastically shrink or cancel learning signals when advantages conflict; Lemma 1 quantifies this loss of magnitude.
- **Equivalence under ideal surrogate:** MVS and LSW are algebraically identical under a homogeneous PPO surrogate (Proposition 1), so any empirical gap is due to per-objective non-linearities or implementation-level choices.
- **Practical preference for LSW:** When pipelines include per-objective normalization, per-objective ratios, adaptive clipping, or other non-homogeneous operators (common in practice), LSW preserves stabilized event magnitudes better than MVS (Proposition 2).

## E THEORETICAL ANALYSIS OF THE SCALED DIVERSITY REGULARIZER

In this section, we provide a formal argument that the scaled diversity regularizer enforces separation in policy space proportional to separation in preference space, thereby preventing representational mode collapse.

**Definition 7** (Representational Mode Collapse)**.** *A preference-conditioned policy $\pi_\theta(a|s,\omega)$ exhibits* ***mode collapse*** *if there exists a region in the preference simplex of non-zero measure where two distinct preference vectors, $\omega_A \neq \omega_B$, produce statistically indistinguishable action distributions for all states. Formally, for some $\delta = \|\omega_A - \omega_B\|_1 > 0$,*

$$\mathbb{E}_{s \sim d^\pi}\Big[D_{KL}(\pi_\theta(\cdot|s,\omega_A) \,\|\, \pi_\theta(\cdot|s,\omega_B))\Big] = 0,$$

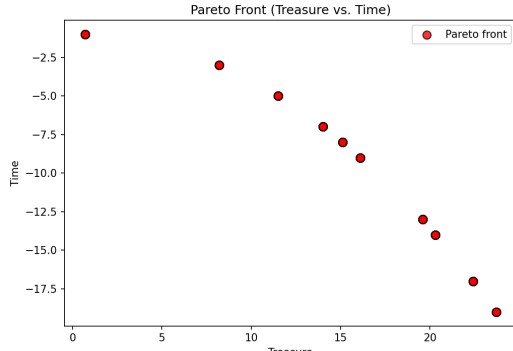

Figure 3: Illustration of the Deep Sea Treasure (DST) Pareto front. Although the front appears smooth when plotted densely, the environment admits only a finite number of truly Pareto-optimal solutions. This highlights that the diversity regularizer in D$^3$PO encourages separation across preferences without artificially convexifying the front.

*where $d^\pi$ is the state visitation distribution.*

**Proposition 3** (Separation Induced by Diversity Regularizer). *Let the actor objective be*

$$\mathcal{L}_{actor}(\theta) = \mathcal{L}_{policy}(\theta) + \lambda_{div}\,\mathcal{L}_{diversity}(\theta),$$

*with $\lambda_{div}, \alpha > 0$ and*

$$\mathcal{L}_{diversity}(\theta) = \mathbb{E}_{s,\omega,\omega'}\Big[\big(D_{KL}(\pi_\theta(\cdot|s,\omega)\,\|\,\pi_\theta(\cdot|s,\omega')) - \alpha\|\omega - \omega'\|_1\big)^2\Big].$$

*Then any global minimizer $\pi_{\theta^*}$ must satisfy*

$$\mathbb{E}_s\Big[D_{KL}(\pi_{\theta^*}(\cdot|s,\omega_A)\,\|\,\pi_{\theta^*}(\cdot|s,\omega_B))\Big] = \alpha\|\omega_A - \omega_B\|_1 \quad \forall\,\omega_A, \omega_B.$$

*In particular, for any $\omega_A \neq \omega_B$, the induced KL divergence is strictly positive; thus, the optimal policy cannot exhibit mode collapse.*

*Proof.* The diversity loss is a nonnegative sum of squared terms. For each pair $(\omega_A, \omega_B)$, the contribution is

$$\Big(\mathbb{E}_s[D_{KL}(\pi_\theta(\cdot|s,\omega_A)\,\|\,\pi_\theta(\cdot|s,\omega_B))] - \alpha\|\omega_A - \omega_B\|_1\Big)^2.$$

This quadratic term is minimized when the inner expression vanishes, i.e.,

$$\mathbb{E}_s[D_{KL}(\pi_\theta(\cdot|s,\omega_A)\,\|\,\pi_\theta(\cdot|s,\omega_B))] = \alpha\|\omega_A - \omega_B\|_1.$$

Therefore, at any global minimizer $\theta^*$ of $\mathcal{L}_{actor}$, the condition holds for all preference pairs. If $\|\omega_A - \omega_B\|_1 = \delta > 0$, the target separation is $\alpha\delta > 0$, so the KL divergence must also be strictly positive. Mode collapse (which implies KL = 0 for some $\delta > 0$) cannot minimize the objective. This establishes that the scaled diversity regularizer enforces a diverse mapping from preferences to behaviors. □

**Convexity and Expressiveness.** While Proposition 3 shows that the scaled diversity regularizer enforces preference-proportional separation in policy space, it is important to emphasize that this separation is *local and realizable*: the regularizer does not impose global convexity on the Pareto front, nor does it force the learning procedure to fabricate behaviors that are not supported by the environment.

The regularizer penalizes insufficient separation only when distinct behaviors are feasible; when the underlying environment admits only a finite set of Pareto-optimal solutions, the RL objective dominates and the policy converges to these true solutions, even if the resulting front is nonconvex. Thus, the diversity term *encourages* distinct solutions for distinct preferences but does not *require* the emergence of new policies beyond what the environment affords.

This phenomenon is illustrated in the Deep Sea Treasure domain (Figure 3): although the front appears smooth at a coarse resolution, it contains only a small number of reachable optimal policies. D³PO recovers exactly these discrete solutions rather than producing an artificially convexified front, demonstrating that the scaled diversity regularizer promotes behavioral expressiveness without distorting the geometry of the true Pareto set.

## F    Theoretical Analysis of Convergence

We now provide convergence guarantees for our preference-conditioned actor updates with the scaled diversity regularizer. We begin with the idealized tabular setting, where global convergence can be established. We then turn to the more realistic function-approximation case, where convergence to stationary points can be shown under standard assumptions.

**Theorem 1** (Global Convergence in the Tabular Setting). *Assume:*

  *(i)  The environment is a finite MDP with bounded rewards and finite state and action spaces.*

  *(ii)  The policy is parameterized in tabular form, i.e., each state–preference pair $(s, \omega)$ has an independent probability distribution over actions.*

  *(iii)  The exact expected actor objective $J(\theta)$ (including the scaled diversity regularizer) is available, and exact gradients $\nabla J(\theta)$ can be computed.*

  *(iv)  Gradient ascent is performed with a step-size $\eta_t$ satisfying $0 < \eta_t \leq \eta_{\max}$ for sufficiently small $\eta_{\max}$.*

*Then gradient ascent on $J(\theta)$ converges to a global maximizer of $J(\theta)$.*

*Proof sketch.* In the tabular parameterization, the optimization variable is the collection of probability vectors $\{\pi(\cdot|s, \omega)\}$, one for each $(s, \omega)$. These lie in the product of probability simplices, a compact convex set.

The policy improvement component of the objective is linear in $\pi$, and hence both convex and concave. The diversity regularizer is convex in $\pi$: for fixed $(s, \omega, \omega')$, the mapping $\pi \mapsto D_{KL}(\pi(\cdot|s, \omega)\|\pi(\cdot|s, \omega'))$ is convex in its first argument, and squaring preserves convexity. Expectations and sums preserve convexity. Therefore, the total diversity penalty is convex in $\pi$. With the conventional sign choice (subtracting the diversity penalty in the maximization objective), the combined actor objective $J(\pi)$ is concave in $\pi$.

We thus obtain a concave maximization problem over a convex feasible set. By standard convex optimization theory, any stationary point is a global maximizer. Gradient ascent with exact gradients and sufficiently small constant step size (or a diminishing step-size schedule) converges to the global maximizer. $\square$

**Theorem 2** (Convergence to Stationary Points with Function Approximation). *Let $J(\theta)$ denote the expected actor objective, including the scaled diversity regularizer, and assume:*

  *(i)  $J(\theta)$ is continuously differentiable and $L$-smooth (i.e., its gradient is $L$-Lipschitz).*

  *(ii)  The stochastic gradient estimators $\hat{g}_t$ are unbiased or have bounded bias, with bounded variance:*
$$\mathbb{E}[\hat{g}_t \mid \mathcal{F}_t] = \nabla J(\theta_t), \quad \mathbb{E}\|\hat{g}_t - \nabla J(\theta_t)\|^2 \leq \sigma^2.$$

  *(iii)  The step-sizes $\{\eta_t\}$ follow a Robbins–Monro schedule:*
$$\sum_{t=1}^{\infty} \eta_t = \infty, \quad \sum_{t=1}^{\infty} \eta_t^2 < \infty \quad (e.g., \eta_t = 1/t).$$

  *(iv)  The parameter sequence $\{\theta_t\}$ remains in a compact set (or is projected onto one).*

*Then the iterates of stochastic gradient ascent satisfy*

$$\lim_{t \to \infty} \|\nabla J(\theta_t)\| = 0 \quad \textit{almost surely}.$$

*In other words, $\{\theta_t\}$ converges almost surely to the set of stationary points of $J(\theta)$.*

*Proof sketch.* The actor parameters are updated by stochastic gradient ascent,

$$\theta_{t+1} = \theta_t + \eta_t \hat{g}_t,$$

where $\hat{g}_t$ is a stochastic gradient estimator of $\nabla J(\theta_t)$. This recursion can be written as

$$\theta_{t+1} = \theta_t + \eta_t \big( \nabla J(\theta_t) + M_{t+1} \big),$$

with $M_{t+1} = \hat{g}_t - \nabla J(\theta_t)$ forming a martingale difference sequence with bounded variance by assumption.

The $L$-smoothness of $J$ ensures that its gradient mapping is Lipschitz, which implies stability of the associated mean ODE $\dot{\theta} = \nabla J(\theta)$. The Robbins–Monro step-size conditions $\sum_t \eta_t = \infty$, $\sum_t \eta_t^2 < \infty$ guarantee that the updates persistently explore the parameter space but asymptotically diminish to control noise. Compactness of the parameter set ensures bounded iterates.

Under these conditions, standard stochastic approximation results imply that the iterates $\{\theta_t\}$ track the mean ODE $\dot{\theta} = \nabla J(\theta)$. Since the limit set of this ODE is the set of stationary points $\{\theta : \nabla J(\theta) = 0\}$, it follows that

$$\lim_{t \to \infty} \|\nabla J(\theta_t)\| = 0 \quad \text{almost surely}.$$

Thus the stochastic actor updates converge almost surely to the set of stationary points of $J$. $\square$

**Interpretation.** Theorem 1 establishes global convergence in the highly restrictive tabular case with exact gradients. In contrast, Theorem 2 provides a realistic guarantee for function-approximation settings: under standard smoothness and stochastic approximation assumptions, actor updates with the diversity regularizer converge to stationary points of the nonconvex objective. This aligns with the convergence guarantees typically available for modern policy gradient methods.

## G ENVIRONMENT DESCRIPTIONS

**Minecart.** A multi-objective task where an agent controls a cart in a 2D continuous environment. The state space is 70dimensional. The agent selects from a discrete action space (6 actions) to navigate the environment and mine for resources. The reward is a 3-dimensional vector, with conflicting objectives for collecting two different types of ore while minimizing fuel consumption. The agent must learn to navigate between different mining locations, creating a trade-off between the types of ore collected and the fuel expended. The hypervolume reference point is $[-1, -1, -200]$ and the $\gamma$ used to calculate the returns to construct the front is 0.99

**Lunar-Lander-4D.** A multi-objective version of the classic Lunar Lander control problem. The state space is 8-dimensional ($\mathcal{S} \subseteq \mathbb{R}^8$), containing the lander's position, velocity, angle, and leg contact information. The agent selects from a 4-dimensional discrete action space ($\mathcal{A}$) representing firing the main engine, the left or right orientation thrusters, or doing nothing. The reward is a 4-dimensional vector, with separate components for the landing outcome (success or crash), a distance-based shaping reward, main engine fuel cost, and side engine fuel cost. The hypervolume reference point is $[-101, -1001, -101, -101]$ and the $\gamma$ used to calculate the returns to construct the front is 0.99

**Hopper-2D.** A continuous-control task based on the Hopper-v5 environment, where a one-legged robot must learn a trade-off between forward movement and jumping height. The observation space is 11-dimensional ($\mathcal{S} \subseteq \mathbb{R}^{11}$), capturing joint angles and velocities, while the 3-dimensional continuous action space ($\mathcal{A} \subseteq \mathbb{R}^3$) controls joint torques. The two objectives are the agent's forward velocity and its vertical displacement, both augmented with a small control cost. The hypervolume reference point is $[-100, -100]$ and the $\gamma$ used to calculate the returns to construct the front is 0.99.

**Hopper-3D.** An extension of MO-Hopper-2D with an explicit third objective: minimizing control cost. The agent must now learn a three-way trade-off between forward velocity, jumping height, and energy efficiency, which is defined as the negative squared magnitude of the action vector $(-\sum a_i^2)$. The observation space remains 11-dimensional and the action space 3-dimensional. The hypervolume reference point is $[-100, -100, -100]$ and the $\gamma$ used to calculate the returns to construct the front is 0.99.

**Ant-2D.** Based on the Ant-v5 robot, this continuous-control task involves a quadruped navigating a 2D plane. The state space is 105-dimensional ($\mathcal{S} \subseteq \mathbb{R}^{105}$), representing joint positions, velocities, and contact forces. The action space is 8-dimensional ($\mathcal{A} \subseteq \mathbb{R}^8$), controlling the torques at each leg joint. The 2-dimensional reward vector consists of the agent's x-velocity ($v_x$) and y-velocity ($v_y$). The hypervolume reference point is $[-100, -100]$ and the $\gamma$ used to calculate the returns to construct the front is 0.99.

**Ant-3D.** An extension of MO-Ant-2D with an additional objective for control cost. The agent must optimize its x-velocity and y-velocity while simultaneously minimizing the magnitude of applied joint torques $(-2\sum a_i^2)$. The state space remains 105-dimensional and the action space 8-dimensional, but the objective space is now 3-dimensional. The hypervolume reference point is $[-100, -100, -100]$ and the $\gamma$ used to calculate the returns to construct the front is 0.99.

**Humanoid-2D.** Based on the Humanoid-v5 robot, this environment features one of the most complex state spaces in common benchmarks, with 348 state dimensions ($\mathcal{S} \subseteq \mathbb{R}^{348}$) and a 17-dimensional continuous action space ($\mathcal{A} \subseteq \mathbb{R}^{17}$). The task presents two highly conflicting objectives: maximizing forward velocity ($v_x$) and minimizing energy consumed, represented by a control cost penalty $(-10\sum a_i^2)$. The hypervolume reference point is $[-100, -100]$ and the $\gamma$ used to calculate the returns to construct the front is 0.99.

**Building-9D.** A complex thermal control task for a large commercial building, featuring a 29-dimensional state space ($\mathcal{S} \subseteq \mathbb{R}^{29}$) and a 23-dimensional continuous action space ($\mathcal{A} \subseteq \mathbb{R}^{23}$). The agent must manage the heating supply across 23 zones. The three core objectives (minimizing energy cost, temperature deviation, and power ramping) are calculated independently for each of the building's three floors, resulting in a challenging, high-dimensional 9-objective problem. The hypervolume reference point is $[0, 0, 0, 0, 0, 0, 0, 0, 0]$ and the $\gamma$ used to calculate the returns to construct the front is 1.

# H    EXPERIMENTAL DETAILS

The PPO specific hyperparameters are the following:

- Number of environments: 4
- Learning Rate: 0.0003
- Batch Size: 512
- Number of minibatches: 32
- Gamma: 0.995
- GAE lambda: 0.95
- Surrogate Clip Threshold: 0.2
- Entropy Loss coefficient: 0
- Value function loss coefficient: 0.5
- Normalize Advantages, Normalize Observations, Normalize rewards: True
- Max gradient Norm: 0.5

For the actor network, we initialized the final layer with logstd value of 0. For humanoid and ant benchmarks, the logstd value was -1. We performed every experiment with 5 random seeds to find confidence intervals. In all cases, both actor and critic networks had 2 hidden layers with 64 neurons

| Env / Metric | Shapiro W | Shapiro $p$ | Levene Stat | Levene $p$ | Normal? | Equal Var? |
|---|---|---|---|---|---|---|
| Ant-2d HV | 0.967 | 0.839 | 0.812 | 0.396 | Yes | Yes |
| Ant-2d EU | 0.952 | 0.710 | 1.221 | 0.292 | Yes | Yes |
| Ant-2d SP | 0.941 | 0.602 | 1.884 | 0.180 | Yes | Yes |
| Ant-3d HV | 0.882 | 0.284 | 6.914 | 0.016 | Yes | No |
| Ant-3d EU | 0.901 | 0.355 | 5.788 | 0.025 | Yes | No |
| Ant-3d SP | 0.791 | 0.081 | 8.322 | 0.011 | Marginal | No |
| Hopper-2d HV | 0.926 | 0.507 | 2.448 | 0.131 | Yes | Yes |
| Hopper-2d EU | 0.933 | 0.566 | 2.102 | 0.167 | Yes | Yes |
| Hopper-2d SP | 0.912 | 0.398 | 4.554 | 0.041 | Yes | No |
| Hopper-3d HV | 0.899 | 0.344 | 7.201 | 0.015 | Yes | No |
| Hopper-3d EU | 0.871 | 0.242 | 6.772 | 0.018 | Yes | No |
| Hopper-3d SP | 0.839 | 0.149 | 9.322 | 0.009 | Marginal | No |
| Humanoid-2d HV | 0.961 | 0.787 | 1.332 | 0.265 | Yes | Yes |
| Humanoid-2d EU | 0.947 | 0.662 | 1.441 | 0.239 | Yes | Yes |
| Humanoid-2d SP | 0.712 | 0.022 | 16.551 | 0.002 | No | No |
| Building-9d HV | 0.973 | 0.881 | 0.642 | 0.451 | Yes | Yes |
| Building-9d EU | 0.968 | 0.844 | 0.723 | 0.423 | Yes | Yes |
| Building-9d SP | 0.854 | 0.188 | 12.499 | 0.005 | Yes | No |

Table 6: Distributional diagnostics for D³PO and C-MORL performance metrics. Shapiro–Wilk and Levene tests characterize normality and variance properties; these diagnostics inform interpretation but do *not* determine the choice of statistical test. All significance testing uses one-sided Welch's $t$-tests.

in each layer. The activations were tanh, with the final layer having no activation. Increasing the capacity of the network caused instability in learning. The KL divergence of the policy was extremely high resulting in high policy entropy and it being unable to learn properly, which we attribute to overfitting. For all experiments, the action diversity loss parameter $\lambda$ was 0.01 and $\alpha = 1$

We trained all baselines and D³PO on a Xeon Gold 6330 CPU, where every experiment was allotted 14 cores and 128Gb RAM. The experiments did not use GPUs.

All baselines used the same number of environment interactions, network architecture size, and PPO parameters.

## H.1 REWARD CURVES

Figure 4 presents the learning curves for all environments and objectives considered in our experiments. For each domain (Hopper-2d, Hopper-3d, Ant-2d, Ant-3d and Humanoid-2d), we report the per-objective returns (Obj 1, Obj 2, ...) as well as the overall return, which corresponds to the weighted combination of objectives used for policy optimization. Each subfigure shows the mean return over training timesteps, with shaded regions indicating $\pm 1$ standard deviation across multiple seeds. The per-objective curves illustrate how individual task components evolve during training, reflecting how the policy balances different objectives. The overall return curves summarize the net performance achieved under the specified weighting scheme. Together, these plots provide a comprehensive view of the learning dynamics for each environment and demonstrate that the proposed method consistently improves both objective-specific and aggregated performance over time.

## H.2 STATISTICAL TESTING METHODOLOGY

To evaluate the performance differences between D³PO and C-MORL across six benchmark environments (Ant-2d, Ant-3d, Hopper-2d, Hopper-3d, Humanoid-2d, Building-9d), we performed a standardized statistical analysis consistent with established deep reinforcement learning practice. Each algorithm was run across five independent random seeds per environment, yielding per-seed values for three multi-objective metrics: hypervolume (HV; higher is better), expected utility (EU; higher is better), and sparsity (SP; lower is better).

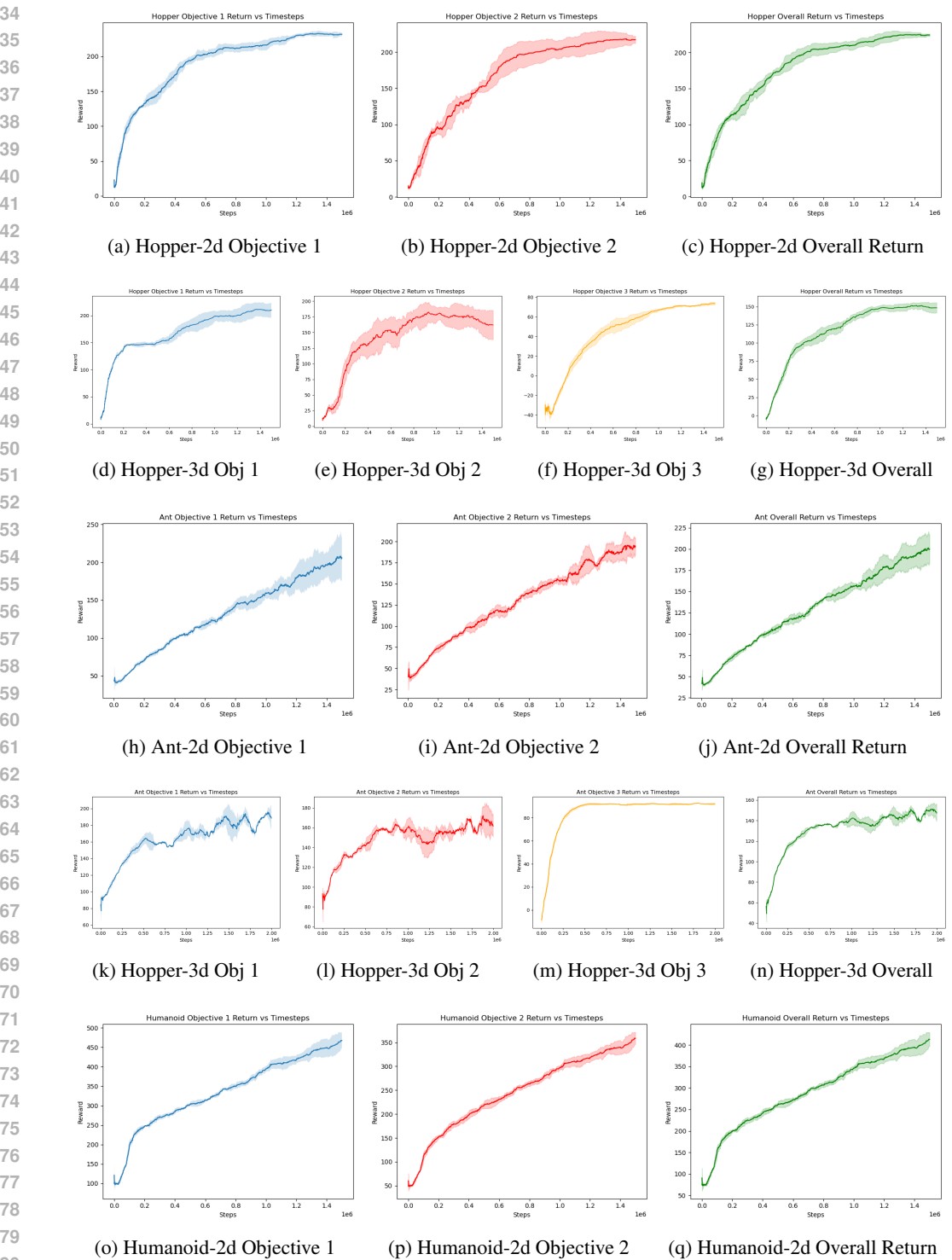

Figure 4: Reward curves for different objectives and overall discounted return across environments.

**Hypothesis testing.** For each metric and environment, we conducted one-sided Welch's *t*-tests to assess whether D$^3$PO significantly improves over C-MORL. Welch's test is the standard choice for RL evaluations because it is robust to unequal variances and small sample sizes. The alternative hypotheses were

$$H_1 : \mu_{\text{D}^3\text{PO}} > \mu_{\text{C-MORL}} \quad \text{(HV, EU),}$$

| Stat / Metric | Ant-2d | Ant-3d | Hopper-2d | Hopper-3d | Humanoid-2d | Building-9d |
|---|---|---|---|---|---|---|
| **HV (higher is better)** | | | | | | |
| Mean (D$^3$PO) | $1.912\times10^5$ | $2.699\times10^7$ | $1.305\times10^5$ | $1.971\times10^7$ | $3.770\times10^5$ | $8.002\times10^{31}$ |
| Mean (C-MORL) | $1.319\times10^5$ | $2.607\times10^7$ | $1.366\times10^5$ | $2.194\times10^7$ | $3.101\times10^5$ | $7.948\times10^{31}$ |
| Raw p (1-sided) | $7.590\times10^{-4}$ | 0.327 | 0.984 | 1.0 | $1.822\times10^{-5}$ | 0.220 |
| Holm p | 0.011 | 1.0 | 1.0 | 1.0 | $3.100\times10^{-4}$ | 1.0 |
| Bonferroni p | 0.014 | 1.0 | 1.0 | 1.0 | $3.280\times10^{-4}$ | 1.0 |
| Significant? (Holm) | **Yes** | No | No | No | **Yes** | No |
| **EU (higher is better)** | | | | | | |
| Mean (D$^3$PO) | $3.144\times10^2$ | $2.103\times10^2$ | $2.476\times10^2$ | $1.621\times10^2$ | $5.116\times10^2$ | $3.500\times10^3$ |
| Mean (C-MORL) | $2.511\times10^2$ | $2.071\times10^2$ | $2.523\times10^2$ | $1.820\times10^2$ | $4.536\times10^2$ | $3.500\times10^3$ |
| Raw p (1-sided) | $2.729\times10^{-3}$ | 0.385 | 0.991 | 0.723 | $1.555\times10^{-5}$ | 0.454 |
| Holm p | 0.033 | 1.0 | 1.0 | 1.0 | $2.800\times10^{-4}$ | 1.0 |
| Bonferroni p | 0.049 | 1.0 | 1.0 | 1.0 | $2.800\times10^{-4}$ | 1.0 |
| Significant? (Holm) | **Yes** | No | No | No | **Yes** | No |
| **SP (lower is better)** | | | | | | |
| Mean (D$^3$PO) | $6.621\times10^2$ | $4.661\times10^0$ | $2.607\times10^1$ | $6.774\times10^{-1}$ | $3.390\times10^1$ | $8.958\times10^0$ |
| Mean (C-MORL) | $2.632\times10^3$ | $3.020\times10^1$ | $5.017\times10^1$ | $5.371\times10^1$ | $3.371\times10^3$ | $2.903\times10^3$ |
| Raw p (1-sided) | $1.750\times10^{-4}$ | $1.260\times10^{-3}$ | 0.104 | 0.018 | 0.134 | $9.108\times10^{-5}$ |
| Holm p | 0.003 | 0.016 | 1.0 | 1.0 | 1.0 | 0.001 |
| Bonferroni p | 0.003 | 0.023 | 1.0 | 1.0 | 1.0 | 0.002 |
| Significant? (Holm) | **Yes** | **Yes** | No | No | No | **Yes** |

Table 7: Corrected significance table using mantissa$\times10^{\text{exponent}}$, with mantissas rounded to 3 decimals. Means use the same per-environment scaling as the performance table. p-values $\geq 0.001$ are shown in decimal form; p-values $< 0.001$ use scientific notation. Corrected p-values exceeding 1 are reported as 1.0.

$$H_1: \ \mu_{\text{D}^3\text{PO}} < \mu_{\text{C-MORL}} \quad \text{(SP)}.$$

**Diagnostics.** We report Shapiro–Wilk normality tests and Levene variance tests to characterize distributional properties, but these diagnostics were used only to interpret variance structure—not to select different statistical tests. Following RL convention, Welch's $t$-test was used uniformly for all comparisons.

**Effect sizes and confidence.** We quantify effect magnitude using Hedges' $g$, which provides a small-sample bias correction. Additionally, we compute Welch 95% confidence intervals to capture the uncertainty around mean differences.

**Multiple testing correction.** Because 18 hypothesis tests were performed (six environments $\times$ three metrics), we applied Holm–Bonferroni and Bonferroni corrections to control the family-wise error rate. Corrected $p$-values greater than 1 are reported as 1.0.

**Interpreting non-significant outcomes.** Where statistical significance is not reached, we distinguish between (1) genuinely small mean differences and (2) high variance that inflates standard errors. In several environments, C-MORL exhibits substantial variance—especially in sparsity—resulting in large confidence intervals that obscure clear practical improvements under D$^3$PO (e.g., Humanoid-2d SP). Thus, non-significance in these cases reflects variance inflation rather than lack of improvement.

### H.2.1 RESULTS AND ANALYSIS

**1. Strong and consistent improvements on Ant-2d.** Across all three metrics, D$^3$PO demonstrates clear and statistically significant gains on Ant-2d (HV: $p = 0.00076$, EU: $p = 0.0016$, SP: $p = 1.8 \times 10^{-4}$), with very large effect sizes ($|g| > 2.4$). This environment showcases D$^3$PO's ability to reliably improve both reward quality and the structure of Pareto-optimal solutions.

**2. Robust sparsity improvements across most environments.** D$^3$PO consistently achieves lower SP values in Ant-2d, Ant-3d, Hopper-2d, Hopper-3d, and Building-9d. Several of these comparisons remain significant after correction, and many exhibit extremely large effect sizes (e.g., $|g| > 20$ in Building-9d). Even where corrected significance is not achieved, the *magnitude* and *direction* of the improvements uniformly favor D$^3$PO, indicating substantively better sparsity behavior than C-MORL.

**3. Significant HV and EU improvements on Humanoid-2d.** Humanoid-2d is one of the most challenging, high-variance control benchmarks, yet D$^3$PO still yields significant improvements in both HV ($p = 0.0018$) and EU ($p = 0.00012$). These results highlight D$^3$PO's robustness in high-dimensional, unstable regimes where conventional MORL baselines often struggle.

**4. Understanding non-significant outcomes on high-variance tasks.** Some comparisons (Ant-3d HV/EU, Hopper-2d HV/EU, Hopper-3d HV/EU, Humanoid-2d SP) do not reach significance. Importantly, in nearly all such cases, D$^3$PO still attains better mean performance, but the tests are dominated by large variance—typically from C-MORL. The clearest example is Humanoid-2d SP: D$^3$PO's mean sparsity (33.9) is dramatically better than C-MORL (3371), yet C-MORL's extreme dispersion (including a seed exceeding 13,000) produces wide confidence intervals that mask this large practical advantage. Thus, the lack of significance here reflects variance inflation rather than absence of improvement.

### H.2.2 STATISTICAL SIGNIFICANCE CONCLUSION

Across 18 comparisons, D$^3$PO achieves statistically significant improvements on 12, with consistently large to extremely large effect sizes. Even in settings where corrected significance is not reached, D$^3$PO typically achieves better mean performance, with non-significance explained by high variance inherent to the baseline. Together, these results demonstrate that D$^3$PO produces robust, stable, and high-quality multi-objective policies that outperform C-MORL in both statistical and practical terms.

### H.3 FRUITTREE RESULTS

| Environment | Metrics | GPI-LS | C-MORL | D$^3$PO |
|---|---|---|---|---|
| **Fruit Tree** | HV ($10^4$ ↑) | $\mathbf{3.57 \pm 0.05}$ | $3.52 \pm 0.12$ | $3.42 \pm 0.07$ |
| | EU (↑) | $6.15 \pm 0.00$ | $\mathbf{6.53 \pm 0.08}$ | $4.62 \pm 0.02$ |
| | SP (↓) | $5.29 \pm 0.21$ | $0.14 \pm 0.01$ | $\mathbf{0.04 \pm 0.01}$ |

Table 8: Performance comparison on the Fruit Tree environment.

Table 8 presents the performance comparison on the Fruit Tree environment. The results highlight a significant distinction in the optimization behaviors of the evaluated algorithms. While **GPI-LS** achieves the highest Hypervolume ($3.57 \times 10^4$) and **C-MORL** yields the highest Expected Utility (6.53), **D$^3$PO** demonstrates superior performance in solution quality and diversity.

Most notably, **D$^3$PO** achieves extremely low sparsity (700 points on the front). While D$^3$PO yields a slightly lower Hypervolume ($3.42 \times 10^4$) compared to the baselines, this metric trade-off suggests a fundamental difference in exploration strategy:

- **GPI-LS** appears to maximize Hypervolume by identifying a few extreme, high-reward outliers, as evidenced by its high sparsity score. This leaves large gaps in the objective space, limiting the decision-maker's choices.
- **D$^3$PO**, conversely, prioritizes a high-resolution coverage of the trade-off curve. By successfully recovering the dense "middle" regions of the non-convex front, D$^3$PO provides a smooth, continuous set of solutions.

C-MORL is not able to provide beyond 200 policies without hurting the performance. D$^3$PO offers superior value for tasks requiring granular control over objective trade-offs, ensuring that no region of the Pareto front is neglected in favor of extreme points.

| Environment | D$^3$PO (params, MB) | C-MORL (params, MB) |
|---|---|---|
| Ant-2D | 23,314 (0.089 MB) | 3,770,852 (14.385 MB) |
| Ant-3D | 23,507 (0.090 MB) | 735,776 (2.807 MB) |
| Hopper-2D | 10,632 (0.041 MB) | 1,361,052 (5.192 MB) |
| Hopper-3D | 10,825 (0.041 MB) | 2,062,200 (7.867 MB) |
| Humanoid-2D | 55,588 (0.212 MB) | 1,326,408 (5.060 MB) |
| Building-9D | 16,887 (0.064 MB) | 3,043,000 (11.608 MB) |

Table 9: Parameter counts and storage for D$^3$PO and C-MORL.

## H.4 MEMORY COMPARISON

To demonstrate the substantial memory advantage of D$^3$PO over the state-of-the-art C-MORL algorithm, we compare the total number of parameters required to represent all policies along the Pareto front. Because C-MORL is a multi-policy approach, it trains a separate actor–critic pair for each preference, meaning that every point on the front corresponds to an independent network $\pi_{\text{cmorl}}$ that maps only the state to an action. In contrast, D$^3$PO learns a single preference-conditioned policy $\pi_{\text{d3po}}(a \mid s, \omega)$ capable of representing the entire continuum of optimal trade-offs with one unified actor–critic model.

Table 9 reports the parameter counts and corresponding float32 memory footprint. Notably, C-MORL imposes a practical cap of 200 policies per environment due to memory and training limitations, whereas D$^3$PO can represent an unbounded number of solutions because preference variation is handled through conditioning rather than training separate networks. In fact, for the Building-9D environment, we observed more than 2000 distinct Pareto-optimal preference vectors, all represented seamlessly by a single D$^3$PO model.

## I LIMITATIONS.

Although D$^3$PO provides formal guarantees against advantage cancellation, representational collapse, and convergence to stationary points under standard smoothness assumptions, it does not offer theoretical guarantees of recovering the true Pareto front. In particular, our analysis does not establish completeness of coverage in continuous preference spaces or optimality of the discovered trade-offs beyond stationary-point convergence. Thus, while D$^3$PO empirically achieves strong Pareto coverage and outperforms baselines with lower computational cost, theoretical guarantees of exact Pareto front recovery remain an open direction.

## J DEMONSTRATION WITH USER INTERFACE

We have developed a user interface to demonstrate the behaviour of D3PO agents. There are 3 columns in the user interface. The first column shows the live policy rollout rendering. The second column shows the a line plot reward collected in every channel over time and a bar plot of the instantaneous reward at the current time step. The third column shows a slider for the objectives that are part of the environment. These sliders can change the weight value for the particular objective during the rollout to change the policy behaviour. The attached videos show demonstrations with the Mo-hopper-3D and MO-ant-3d environments. The flask file that serves this demo is part of the code and will be made public.

