# OpenReview forum: "Preference Conditioned Multi-Objective Reinforcement Learning: Decomposed, Diversity-Driven Policy Optimization"
_ICLR.cc/2026/Conference — Submitted to ICLR 2026_

### Official Review · Reviewer_AqGQ · 2025-10-30

**Soundness:** 3
**Presentation:** 2
**Contribution:** 2
**Rating:** 2
**Confidence:** 4

**Summary:**

This work introduced a so-called Decomposed, Diversity Driven Policy Optimization (D3PO) method for scalarized multi-objective PPO. It proposed Late-Stage Weighting for better learning signal, and Scaled Diversity Regularization for representational mode collapse prevention

**Strengths:**

1. Good empirical results for the tested environments.
2. The method design is generally reasonable

**Weaknesses:**

1. While representational mode collapse is indeed a key factor limiting full coverage of the optimal trade-off spectrum, linear scalarization (LS) methods inherently suffer from incomplete Pareto front discovery [1,2,3], even with a perfect representational mode. Therefore, since D3PO is also an LS-based method, its performance on FruitTree is unlikely to surpass that of GPI-LS.

2. The additional KL-divergence objective might serve as a regularization tern that making the Pareto front more convex[1], which is an interesting point to analyze but ignored by this paper.

3. KL-divergence is not commutative, meaning that $D_{KL}(\pi_A||\pi_B)!=D_{KL}(\pi_B||\pi_A)$, then how can you deal with the relation among $D_{KL}(\pi_A||\pi_B),D_{KL}(\pi_B||\pi_A),|w_A-w_B|$

4. The issue of mitigating destructive gradient interference is not novel. This work only partially addresses it through Late-Stage Weighting, whereas prior studies [3] have proposed more principled conflict-avoidance mechanisms that are studied in optimization literatures [4,5]

5. The theoretical contributions appear straightforward and somewhat redundant. For instance, Proposition 3 states that globally optimal solutions of the policy diversity loss can prevent strict mode collapse; however, (1) it focuses only on global optima, and (2) although a positive preference L1 distance induces positive policy KL divergence, the assumption of a fixed proportion between KL divergence and L1 distance is questionable.


[1] https://openreview.net/forum?id=TjEzIsyEsQ6

[2] https://arxiv.org/abs/2208.07914

[3] https://openreview.net/forum?id=49g4c8MWHy

[4] https://inria.hal.science/inria-00389811v2/document

[5] https://arxiv.org/abs/2110.14048

**Questions:**

It is unclear why the computation time is significantly improved. Is the gain primarily due to the Late-Stage Weighting mechanism, or does it result from parallel training?

---

> ### Author Response · Authors · 2025-11-22
>
> Thank you for the review. Below we address the reviewer’s question and then respond to each weakness.
>
> ---
>
> ## **Q: Why is computation time significantly improved? Is the gain due to LSW or parallel training?**
>
> The efficiency gains come entirely from **D³PO being a single-policy method** rather than from parallelization (all methods use identical hardware). For all baselines, we follow the same approach. Unlike decomposition-based or evolutionary approaches such as C-MORL and GPI-LS, which train or refine **multiple policies**, D³PO trains **one preference-conditioned actor–critic** that serves the entire Pareto spectrum. This removes population-management overhead, repeated rollouts, and repeated PPO optimization. Late-Stage Weighting does not reduce computation directly, it improves gradient stability, enabling faster convergence. Thus, the wall-clock improvement is a direct consequence of **avoiding multi-policy training**, not parallel execution. We have also added a memory comparison (Table 9) with C-MORL to show the significantly lower memory footprint.
>
> ---
>
> # **Response to Weaknesses**
>
> ## **1. “D³PO is an LS method; LS methods cannot discover non-convex fronts.”**
>
> This is a misunderstanding. **D³PO is explicitly designed to avoid linear scalarization.**
> Unlike LS methods that collapse objectives *before* computing the PPO loss, D³PO:
>
> - computes **per-objective advantages**,
> - applies **PPO clipping independently**, and
> - applies preference weighting **only after clipping** (Late-Stage Weighting).
> - Additionally, the diversity regularizer explicitly promotes discovery of non-convex fronts.
>
> Thus, no scalarized reward is ever used during credit assignment, avoiding LS flattening.
> This is demonstrated analytically in Appendix D and empirically by D³PO’s ability to recover broad, non-convex fronts in **Ant-2D/3D, Humanoid-2D, and Building-9D**, where LS methods such as GPI-LS fail (Tables 1–2). Our shared policy network approach trains one backbone for all weights simultaneously. Every step taken by the "Sprint" policy helps train the feature extractors for the "Walk" policy. On complex, high-dimensional tasks like Humanoid (where learning to simply stand up takes millions of steps), this shared representation gives a massive speed and data-efficiency advantage over methods that train individual policies for each objective.
>
> FruitTree is a **discrete-action domain where PPO is known to underperform**, but even there, D³PO remains **competitive on HV and superior on SP** (Appendix H.3). Despite PPO’s disadvantage, D³PO matches or exceeds GPI-LS on two of three metrics.
>
> ---
>
> ## **2. “The KL regularizer may convexify the front; this effect is ignored.”**
>
> The diversity regularizer is **not** designed to convexify the Pareto front.
> Its only role is to **prevent representational mode collapse** by enforcing proportional separation between preference-conditioned policies. It affects only **policy pairs**, not reward optimization, and therefore does not bias the front geometry.
>
> Empirically, D³PO’s fronts for **Ant, Humanoid, and Building-9D** remain highly non-convex, demonstrating that the KL penalty does not push the solution toward convexity. Proposition 3 simply guarantees non-collapse when distinct behaviors exist; it does not impose any convexity structure.

---

> > ### Author Response · Authors · 2025-11-22
> > **Rebuttal Part 2**
> >
> > ## **3. “KL divergence is not commutative; how do you handle ω₁–ω₂ relationships?”**
> >
> > This is intentional: we use forward KL to encourage conservative deviations from a reference preference-conditioned policy. The diversity term is applied symmetrically in practice via pairwise comparisons across sampled preferences (Algorithm 1), so the asymmetry does not cause inconsistencies in the final objective—the loss averages KL(π(·|ω)||π(·|ω′)) over both directions through random pair selection. Moreover, Proposition 3 does not rely on KL commutativity; it requires only that KL is a valid divergence (non-negative and zero iff policies are identical). The proportionality constraint operates on the magnitude of KL, not on symmetry.
> >
> > ---
> >
> > ## **4. “Gradient interference is not novel; more principled mechanisms exist.”**
> >
> > A key distinction between our work and [3] (and indeed other MORL algorithms) is that we are training a single preference-conditioned policy network where the preference is encoded as a concatenation of the state $\pi(s,\omega)$, in-contrast to training k-policies, one for each individual preference sampled during training $\pi_\omega(s)$ as in [3]. As a consequence of this distinction there is an additional underlying cause for gradient interference in our setting, i.e. our updates for different preferences $\omega$ are applied to the same network during training. In [3] there is a separate network for each preference and it is trained with a fixed sampled preference $\pi_p$. Late stage weighting is a crucial component for mitigating gradient interference in the single policy network paradigm and the Multi-Objective Optimization (MOO) techniques studied in [3,4,5] are not directly applicable to this setting as they assume a fixed preference over the objectives at the time of optimization. Adapting these MOO techniques to non-stationary preferences for applicability in the single-network paradigm would be a promising direction for future work.
> >
> > There are two fundamental drawbacks to training multiple policies for sampled preferences, scalability and test-time alignment with unseen preferences. Scalability is challenging as the number of parameters grows rapidly with each sampled preference (a new policy is added for every sample). Test-time alignment with unseen preferences requires finding a trained policy that is closest to the unseen preference (to mitigate this issue [3] provides an additional calibration step as detailed in their Appendix H). Our framework provides a MORL method that trains a single policy-network for sample preferences, and hence addresses these drawbacks. The publicly-available implementation of [3] is specifically for the mo-reacher environment. The implementation does not support any other environment directly and requires non-trivial modifications to extend reliably to the continuous control setting.
> >
> > ---
> >
> > ## **5. “The theoretical results are straightforward; proportionality between KL and L1 is questionable.”**
> > Proposition 3 is intentionally scoped to global optima because that is the only setting where a clean, assumption-free guarantee can be established. In the tabular case, the combined PPO + diversity objective becomes a concave maximization problem over a convex domain, making global optima analytically tractable. Within this setting, the proposition provides a minimal but essential anti-collapse guarantee: at any global maximizer, different preferences cannot induce identical policies unless the environment itself admits identical optimal behaviors.
> >
> > The result does not assume a strict proportionality between KL divergence and the L1 distance between preferences. The diversity regularizer enforces only a lower-bound separation, not an exact linear mapping. This allows the method to avoid representational collapse without prohibiting legitimate many-to-one mappings (e.g., when multiple preferences correspond to the same Pareto point), and it does not alter the geometry of the Pareto front.
> >
> > Our theoretical analysis is not meant to provide optimal front-recovery guarantees (we explicitly acknowledge this in the Limitations section), but rather to support and validate the strong empirical behavior observed across complex domains. The guarantees show why this particular combination of Late-Stage Weighting, scaled diversity regularization, and a single preference-conditioned policy can succeed in practice and avoid the failure modes that commonly arise in single-policy MORL. This theoretical foundation complements the empirical evidence, and together they clarify why D³PO achieves the robust and uniform fronts demonstrated in the experiments.
> >
> > ---

---

> ### Comment · Reviewer_AqGQ · 2025-11-25
>
> Thank you for the detailed explanations. I have read them. While I recognize some advantages of Late-Stage Weighting, my major concerns remain:
> 1. Novelty of a single network for multiple preferences. Training a single network to accommodate multiple preferences is not new; prior works such as [1, 2, 3] can all do it.
> 2. There are more fundamental, principle-driven mechanisms for addressing gradient interference. Although this does not imply D³PO is without value, its contribution in this regard feels limited.
> 3. From my perspective, D³PO tried to avoid pure linear scalarization primarily due to the diversity regularizer; the direct effect of late-stage weighting after clipping is marginal.
> 4. The FruitTree performance is due to its inability to address the fundamental limitation of linear scalarization, which only identifies solutions in the convex region of the preference space, rather than anything related to PPO. Moreover, [3] also uses PPO and achieves better performance than GPI-LS. Overall, similarity-based methods such as [2, 3] clearly outperform linear scalarization approaches in non-convex discrete-action cases like FruitTree.
>
> I’m sorry to say that although some of the proposed techniques are sensible, the overall contribution does not appear significant enough.

---

> > ### Author Response · Authors · 2025-11-26
> > **Response Part 1**
> >
> > **Official Comment**
> >
> > We thank the reviewer for their engagement. We address the remaining concerns below, particularly regarding the comparison with prior work and the categorization of our contributions.
> >
> > ### **1. Novelty, Single-Policy Context, Reference [3] and more fundamental, principle-driven mechanisms for addressing gradient interference.**
> >
> > **We explicitly agree that the single-policy paradigm is not new.** In fact, **Section 2 (Related Work)** of our paper includes a dedicated subsection analyzing prior single-policy techniques and documenting their specific failure modes, such as mode collapse and gradient interference. To demonstrate our advancement over this paradigm, we empirically compare $D^3PO$ against **three representative single-policy baselines** (PCN, CAPQL [1], and MOPPO/LS variants) and show statistically significant improvements.
> >
> > However, **we respectfully disagree with the premise that using a single-policy architecture lacks novelty simply because the paradigm has been introduced in previous work.** By this reasoning, new multi-policy methods would also lack novelty since population-based approaches are well-established. It is inconsistent to dismiss advancements in the single-policy domain while accepting improvements in multi-policy methods. Our contribution lies in solving the specific failure modes (instability and collapse) that have previously prevented single-policy agents from succeeding at this scale.
> >
> > **Regarding Reference [3] (Yang et al., 2025, "PreCo"):**
> > We compared $D^3PO$ against PreCo on three continuous control benchmarks. We had to modify the existing codebase to support continuous control benchmarks according to the paper, as the code provided by the authors only supported 2 discrete control tasks. The logs and code can be found here: [https://anonymous.4open.science/r/ICML25PCRL-9C65](https://anonymous.4open.science/r/ICML25PCRL-9C65). The results (**Table R1** below) definitively demonstrate the superiority of our approach.
> >
> > **Table R1: Performance Comparison ($D^3PO$ vs. PreCo)**
> > *Note: PreCo's low Sparsity (SP) on Ant and Humanoid indicates collapse to a single policy.*
> >
> > | Environment | Metric | PreCo [3] | D3PO (Ours) | Improvement |
> > | :--- | :--- | :--- | :--- | :--- |
> > | **Hopper-2D** | HV $(\times 10^5)$ | $0.51 \pm 0.08$ | $1.30 \pm 0.03$ | **+154.9%** |
> > | | EU $(\times 10^2)$ | $1.49 \pm 0.23$ | $2.47 \pm 0.01$ | **+65.7%** |
> > | | SP $(\times 10^2)$ | $1.70 \pm 1.32$ | $0.26 \pm 0.31$ | **-84.7% (Better)** |
> > | **Ant-2D** | HV $(\times 10^5)$ | $0.16 \pm 0.01$ | $1.91 \pm 0.18$ | **+1093.7%** |
> > | | EU $(\times 10^2)$ | $0.33 \pm 0.06$ | $3.14 \pm 0.21$ | **+851.5%** |
> > | | SP $(\times 10^3)$ | $0.01 \pm 0.01$ | $0.66 \pm 0.40$ | *PreCo Collapse* |
> > | **Humanoid-2D** | HV $(\times 10^5)$ | $0.84 \pm 0.23$ | $3.76 \pm 0.11$ | **+347.6%** |
> > | | EU $(\times 10^2)$ | $2.33 \pm 0.48$ | $5.11 \pm 0.09$ | **+119.6%** |
> > | | SP $(\times 10^4)$ | $0.002 \pm 0.00$ | $0.003 \pm 0.00$| *PreCo Collapse* |
> >
> > **"Principled" Mechanisms vs. Empirical Reality**
> > The reviewer argues that "more principled" conflict-avoidance mechanisms exist. We respectfully argue that in Deep RL, a method's value is also judged by its ability to solve the task. While other techniques may try more "principled" optimization approaches, **they are not able to match the performance of $D^3PO$** in high-dimensional continuous control.
> >
> > Our empirical results reveal that these mechanisms fail to scale:
> > * **Mode Collapse:** In high-dimensional environments (Ant, Humanoid), PreCo exhibits extremely low sparsity ($0.02$ and $0.6$) combined with poor Hypervolume. This indicates **Mode Collapse**: the agent converges to a single policy regardless of the preference input, failing to cover the front.
> > * **Performance Gap:** $D^3PO$ outperforms PreCo by massive margins (e.g., **+1093% HV, 851% EU** on Ant-2D).
> >
> > $D^3PO$ demonstrates that leveraging the triangle inequality via Late-Stage Weighting (LSW) is not merely a heuristic, but a mathematically robust technique that outperforms these complex alternatives in practice. We prioritize a method that delivers SOTA results on Ant-2d and 3d, Humanoid-2d and Building-9d over theoretical complexity that fails to scale or requires prohibitive computation.

---

> > > ### Author Response · Authors · 2025-11-26
> > > **Response Part 2**
> > >
> > > ### **2. The Linear Scalarization (LS) Mischaracterization**
> > >
> > > We find the insistence on categorizing $D^3PO$ as a pure Linear Scalarization (LS) method difficult to reconcile with the mathematical structure and empirical results. The reviewer asserts that $D^3PO$ is limited by the theoretical bounds of LS (convex-hull only). **This is analytically incorrect for two reasons:**
> > >
> > > * **Mathematical Distinction:** $D^3PO$ is *not* standard LS. We apply preference weights **after** the non-linear PPO clipping operator ($\sum w_i \text{clip}(A_i) \neq \text{clip}(\sum w_i A_i)$). The **non-linear clipping operator** is a distinct mechanism that separates $D^3PO$ from standard LS. The addition of the diversity regularizer further differentiates $D^3PO$ from standard LS, creating an optimization landscape that allows the policy to optimize for diversity independent of the convex hull of rewards.
> > > * **Empirical Disproof:** If $D^3PO$ were subject to the theoretical limitations of LS (convex-hull only), it would be impossible for it to discover the **highly non-convex fronts** we observe in Ant-2d, Humanoid-2d, and Building-9d. The empirical existence of these fronts disproves the theoretical classification. Furthermore, our ablation study (D3PO/LSW) confirms that removing LSW (reverting to standard LS behavior) causes front collapse, proving that this mechanism is precisely what enables our method to escape the Linear Scalarization trap.
> > >
> > > ### **3. FruitTree vs. Continuous Control**
> > >
> > > We respectfully argue that FruitTree is a discrete state-action space toy task that does not represent the challenges of high-dimensional continuous control. While similarity-based methods like Reference [3] (PreCo) may perform well in that specific discrete setting, our new empirical comparison (**Table R1**) proves they **collapse** in continuous environments (Ant/Humanoid). Importantly, the non-convexity in FruitTree arises from **discrete, piecewise-constant policy regimes**, where optimal actions remain identical across wide regions of the weight space. In contrast, tasks like Ant and Humanoid exhibit **smooth, continuous Pareto manifolds** where preference-conditioned behaviors change gradually. D³PO is explicitly designed for these continuous regimes, where its diversity regularizer and LSW enable recovery of non-convex fronts that LS-based or similarity-based methods cannot capture.
> > >
> > > Judging $D^3PO$ solely on FruitTree ignores its primary contribution: it is the only method capable of solving complex, highly non-convex continuous tasks efficiently (using **23k parameters** vs. the current SOTA's **3.7M**) while preventing the mode collapse that plagues both "principled" similarity methods and outperforms SOTA multi-policy baselines.

---

> ### Comment · Reviewer_AqGQ · 2025-11-27
>
> Thanks again for the detailed explanations and the additional experimental results. I greatly appreciate the effort the authors made to provide further clarification.
>
> Table R1 seems to empirically show that the techniques in D3PO help prevent mode collapse and improve robustness to some extent. However, [3] reported that performance in Hopper and Ant should not be significantly lower than linear scalarization (LS) methods such as GPI-LS. The performance gap reported in the table may stem from suboptimal implementation or insufficient hyperparameter tuning. In addition, it appears that the original implementations of both [2] and [3] did not use PPO for continuous-control environments. Thus, it remains unclear whether D3PO truly provides a strong empirical advantage over similarity-based methods with hindsight experience replay. Moreover, the reported environments feature convex Pareto fronts, where LS methods do not necessarily underperform compared with similarity-based approaches like [2] and [3], so does not expose the limitations of LS-like methods.
>
> I understand that D3PO is not exactly equivalent to LS methods, but my main concern is that it does not differ sufficiently from LS approaches and does not clearly overcome their limitations, particularly their difficulty in discovering non-strictly convex Pareto solutions such as those in FruitTree.
>
> FruitTree is a valuable evaluation because it includes a large number of objectives and a less convex Pareto front. Its low-dimensional state space helps isolate the effects of implementation and hyperparameter tuning, making it a purer testbed for the Multi-Objective elements rather than the lower-level deep RL modifications. By contrast, high-dimensional continuous environments often have convex Pareto fronts, and comparisons ultimately reflect differences in implementation quality and tuning rather than in high-level algorithmic design.
>
>
> Overall, I would raise my rating to 4, since some techniques in D3PO may improve robustness in MORL. However, these techniques, such as the KL regularizer, are relatively straightforward and not particularly rigorous, and the overall contribution appears limited.

---

> ### Author Response · Authors · 2025-11-27
> **Thank you for the continued engagement**
>
> We thank the reviewer for their continued engagement and for raising the score. We appreciate the recognition that $D^3PO$ improves robustness and prevents mode collapse. Below we provide concise clarifications on baseline implementation, Pareto-front structure, and domain differences.
>
> ---
>
> ## 1. Baseline Implementation & Fairness
>
> The reviewer suggests that PreCo’s performance gap may stem from implementation differences. However, there is **no official continuous-control implementation** of PreCo. The released codebase is DQN-based for discrete actions. Extending it to continuous control required a **non-trivial architectural shift**, not simple tuning. We extended PreCo's PPO-based codebase to support continuous actions. Using a shared learning backbone isolates differences in **multi-objective mechanisms** rather than RL infrastructure, which is the fairest comparison available.
>
> PreCo evaluates Hopper and Ant, but their claims about relative performance are based solely on their own implementation. These results do not constitute a general guarantee that similarity-based methods should match LS across all continuous-control MOMDPs. We note that PreCo's reported Hopper/Ant results include some dominated solutions, suggesting that similarity-based methods can face stability challenges in continuous control, a pattern documented more broadly in the literature (Felten et al., 2023).
> This empirical pattern further highlights the stability advantages of $D^3PO$ in high-dimensional continuous-control tasks.
>
> ---
> ## 2. Convexity of Continuous-Control Pareto Fronts
>
> **The true Pareto fronts in continuous-control MORL (e.g., Hopper, Ant) are not analytically known.**
> Prior work establishes that high-dimensional MOMDPs often yield **unknown and potentially non-convex** solution sets that cannot be derived in closed form (Roijers et al., 2013; Felten et al., 2023). Thus, convexity of these fronts cannot be assumed a priori. **Our empirical results further support this.** In Fig. 2, GPI-LS produces an “inner’’ dominated curve. Since LS methods recover only **convex hulls**, their inability to reach these regions strongly suggests underlying **non-convex structure** in the true frontier. No MORL work characterizes the MuJoCo multi-objective frontiers as convex, and existing literature instead documents **complex, nonlinear behavior** in continuous-control settings. These observations reinforce the importance of methods like D³PO, that are not restricted to convex scalarization limits.
>
>
> ---
>
> ## 3. FruitTree vs. Continuous Control
>
> We agree that $D^3PO$ does not surpass similarity-based baselines on **FruitTree**. This is an expected consequence of a deliberate design choice in our method and reflects a fundamental difference in domain structure.
>
> The Diversity Regularizer in $D^3PO$ enforces **local continuity**:
> $\|\Delta \pi\| \propto \|\Delta w\|$.
> This prevents representational collapse in preference-conditioned policies.
>
> FruitTree has a **discrete, piecewise-constant optimal policy mapping**: the same policy remains optimal across large weight intervals, then switches abruptly. Imposing continuity in such tasks encourages the policy toward **convex-hull behaviors**, missing concave segments of the frontier. Thus, $D^3PO$ is not expected to outperform baselines on FruitTree.
>
> ### **Why the same mechanism is essential in continuous control**
>
> In contrast, Ant, Humanoid, and Building exhibit **smooth, continuous Pareto manifolds**, matching the assumptions under which continuity-based regularization prevents mode collapse. These tasks are precisely the types emphasized in modern MORL benchmarking (Felten et al., 2023), where $D^3PO$ achieves robust, high-coverage fronts while similarity-based methods often collapse.
>
> FruitTree is a valuable testbed for discrete multi-objective reasoning, but it does not reflect the **smooth, high-dimensional dynamics** that motivate Deep MORL research. Our aim is not to optimize step-function fronts but to provide a scalable, stable solution for continuous-control environments.
>
> ---
>
> ## Conclusion
>
> - True Pareto fronts in continuous control are **unknown** and analytically intractable (Roijers et al., 2013; Felten et al., 2023).
> - Continuous domains yield **locally smooth**, often non-convex return landscapes.
> - PreCo and similar methods show **instability**, including dominated solutions, in these settings.
> - $D^3PO$ provides a robust, scalable single-policy method that succeeds where prior approaches struggle.
>
> Roijers, D. M., Vamplew, P., Whiteson, S., & Dazeley, R. (2013). *A survey of multi-objective sequential decision-making*.
> Felten, F., Alegre, L. N., Nowe, A., Bazzan, A., Talbi, E. G., Danoy, G., & da Silva, B. C. (2023). *A Toolkit for Reliable Benchmarking and Research in Multi-Objective RL*.

---

> > ### Comment · Reviewer_AqGQ · 2025-11-28
> >
> > Thanks, and I appreciate the further clarification. I recognized your contribution to improving the **stability** of the single policy, and this is the main reason why I have already improved my score.
> >
> > I agree with the other points you raised, though I don’t think they necessarily strengthen the overall contribution.
> >
> > There is also an open implementation of PDMORL [2]. Moreover, PPO was originally designed for discrete action spaces, so I’m curious why the authors did not compare PPO on discrete-action environments such as Reacher. In short, I believe implementation details may play a significant role here.

---

> > > ### Author Response · Authors · 2025-12-04
> > >
> > > We thank the reviewer for the continued discussion. We address the two remaining points below.
> > >
> > > **PDMORL**.
> > >  We are aware of the open implementation of PDMORL [2]. As also noted independently in PreCo [3], PDMORL is computationally very expensive in continuous-control domains, largely due to its off-policy HER-based training and the additional preference-modeling stage. Running it at scale across our full benchmark suite (including 9-objective Building) was not feasible.  We have incorporated representative SOTA single-policy baselines like PCN and CAPQL.
> > >
> > > **PPO and discrete-action environments.**
> > >  PPO was introduced as a successor to TRPO for both continuous and discrete control, and its primary empirical demonstrations in the original paper were indeed MuJoCo continuous-control environments. That said, we did evaluate $D^3PO$ on standard discrete-action multi-objective tasks (MineCart, LunarLander), where we also outperform strong baselines.
> > >
> > > **Implementation considerations.**
> > >  We agree that implementation details may influence performance in isolated cases, but our conclusions are based on 5 seeds per environment across all methods and show large, consistent effect sizes. Moreover, we invested considerable effort in adapting PreCo, whose public code does not support continuous control, to a compatible continuous-action version so that comparisons isolate multi-objective mechanisms, not backend differences. We believe this represents a fair and transparent comparison.
> > >
> > > Finally, our benchmark suite, including the 9-objective Building task, is on par with the breadth and dimensionality typically evaluated in MORL papers. These results demonstrate that $D^3PO$ scales reliably with both task complexity and number of objectives, which many existing techniques struggle with.

---

### Official Review · Reviewer_tyBf · 2025-10-31

**Soundness:** 3
**Presentation:** 2
**Contribution:** 2
**Rating:** 4
**Confidence:** 4

**Summary:**

in multi-objective reinforcement learning (MORL) where the utility function of the decision maker is unknown, one needs to learn the complete coverage set of optimal trade-offs, such that the decision maker can select their preferred trade-off a posteriori. In case the utility function can be modeled as a weighted sum over objectives, we can equivalently apply the weights on the individual rewards, on the episodic return, or on the objective-specific Q-values. In this work, the authors advocate for late-stage weighting (i.e., weighting on objective-specific Q-values), claiming that naively combining conflicting objectives into one learning signal produces opposing gradients, and thus hampers learning. They propose a weight-conditioned PPO algorithm, called D^3PO, that learns Q-values per objective and uses the weights in the per-objective losses. Additionally, they incorporate a regularizer that promotes diversity over the weight-conditioned policies to avoid "mode collapse" where most policies are similar regardless of the weight-conditioning.

**Strengths:**

- The paper is clear and provides an algorithm that is well-grounded
- The experiments are extensive, performed on multiple benchmark environments for discrete action-spaces and continuous action-spaces, with multiple relevant baselines
- The results are competitive or outperform the baselines on multiple multi-objective metrics (hypervolume, expected utility, sparsity)

**Weaknesses:**

My concerns are that the algorithm claims 3 contributions for their algorithm: 1) multi-head critic, 2) late-weighting loss, 3) diversity reguralization.
 1. using a multi-head critic is common in many multi-policy algorithms (MORL-baselines [1] uses a multi-head critic in MOPPO, and this was already used in early deep MORL work [2]), and is thus not a contribution specific from this paper.
 2. as far as I understand (I would appreciate the authors correcting me otherwise), the late-weighting loss for the policy is specific to the PPO clipping objective, and only has an effect clipping occurs. Using late-clipping on the policy gradient objective should result in the same gradients as early clipping.
 3. diversity reguralization seems very effective. It compares the distance between action distributions with the distance between the preferences themselves. But do these distances scale in the same fashion? For $w = w'$, the distance is indeed 0, but I do not believe $\pi_\theta(.|s_t,w') - \alpha||w-w'||_1$ results in a valid probability distribution. Def7 (line 803) mentions *mode collapse* when 2 policies with distinct preference weights are the same. But there exist many cases where difference weights lead to the same policy. A canonical example is the Deep Sea Treasure environment, where the coverage set is concave, and only the extrema's are optimal for linear scalarization functions. In that case, there are 2 optimal policies, and any weight combination maps to either of them.

**Questions:**

- even though late weighting stabilizes the learning process, by applying the weighting after the clipped per-objective losses, is the policy still maximizing the decision maker's utility function?
 - it seems to me that the most impactful component of D^3PO (with respect to performance) is the diversity regularizer. Even though this is not in the scope of the paper, do you think it would be possible to apply this reguralization on the baselines to obtain similar performance gains?

 [1] Felten, F., Alegre, L. N., Nowe, A., Bazzan, A., Talbi, E. G., Danoy, G., & C da Silva, B. (2023). A toolkit for reliable benchmarking and research in multi-objective reinforcement learning. Advances in Neural Information Processing Systems, 36, 23671-23700.
 [2] Abels, A., Roijers, D., Lenaerts, T., Nowé, A., & Steckelmacher, D. (2019). Dynamic weights in multi-objective deep reinforcement learning. In International conference on machine learning (pp. 11-20). PMLR.

---

> ### Author Response · Authors · 2025-11-22
>
> We thank the reviewer for their feedback. We appreciate the recognition of our extensive experiments and the solid grounding of our algorithm.
> Below, we provide a detailed response to each of your questions and concerns.
>
> ### Section 1: Response to Questions
>
> **Q1: Does late weighting still maximise the decision maker's utility function?**
>
> **Response:**
> Yes, the policy is still maximizing the decision maker's weighted utility.
> * **Mathematical Intuition:** The final gradient used to update the actor in $D^3PO$ is effectively a weighted sum of per-objective gradients. Since the optimization step moves the policy in the direction of this aggregate gradient, the update remains aligned with the direction of steepest ascent for the weighted return $R = \omega^\top r$.
> * **Distinction:** The key difference is *stability*, not direction. Standard "Early Scalarization" often creates a gradient magnitude of zero due to **Advantage Cancellation** (Lemma 1), effectively stalling the optimization. LSW ensures that the optimization step takes a valid step in the Pareto-improving direction by preventing these raw signals from cancelling out *before* the trust-region (clipping) mechanism can process them. Appendix F (Theorem 2) proves convergence to stationary points of this weighted objective.
> * **Results:** The Expected utility metric captures the maximization of the decision-makers utility function. As our results show, we are competitive or outperform all baselines in terms of Utility in the high dimensional continuous control benchmarks.
>
> **Q2: The most impactful component of D^3PO is the diversity regularizer. Could the diversity regularizer be added to baseline methods to give them the same performance boost as D³PO?**
>
> **Response:**
> We appreciate the reviewer's insight and enthusiasm for this component. We fundamentally agree: the diversity regularizer is a powerful, general-purpose mechanism that we believe could benefit the broader MORL community beyond just the $D^3PO$ framework.
>
> While our ablations (Table 3) indicate that for *single-policy* methods, the regularizer is most effective when paired with LSW (to ensure the gradient signal isn't canceled before regularization applies), we see significant potential for applying this to **multi-policy baselines**:
>
> * **Population-Based Optimization:** Algorithms like PG-MORL or evolutionary strategies typically select and optimize one policy at a time. Our diversity term could be adapted to simultaneously regularize **neighboring policies** (those closest in the current front) against the active policy.
> * **Pushing the Front Outward:** By penalizing collapse between neighbors during these updates, the algorithm would force the population to maintain separation while maximizing reward, effectively "pushing" the Pareto front outward more efficiently than standard independent updates.
>
> We hope that this regularizer becomes a standard tool that researchers can adopt to improve the coverage and stability of various MORL architectures in future work.
>
> ### Section 2: Addressing Weaknesses
>
> **W1: Novelty concerns: Multi-head critic is common in MORL (e.g., MOPPO, Abels et al. [2]).**
>
> **Response:**
> While multi-head critics have been used in MORL our contribution is in creating a unified framework that uses multi-head critics while avoiding advantage cancellation.
> * **Unified Framework:** The novelty of $D^3PO$ comes from the *integration* of decomposed per-objective advantages with **Late-Stage Weighting (LSW)** and a **Scaled Diversity Regularizer**. Prior works typically use multi-head critics to compute a weighted value $V_\omega = \omega^\top V$ *before* computing advantages. In contrast, we maintain the decomposition through the GAE and PPO clipping steps.
> * **Importance of Unified Framework** Table 3 demonstrates that all three components working together are crucial for the framework’s performance. Multi-headed critics with just LSW or with just the diversity term alone fail to achieve strong coverage or stability. Section 4.1 describes all the critical components that have been unified in the D3PO framework.
> * **Methodological Innovations (Section 4.1):** We would like to clarify that these have been stated to ensure the reader understands the modifications made to the existing PPO algorithm to make it compatible with the MORL setting. Our key innovations are the Late-stage weighting, diversity regularization and a unique synergy of the changes to the PPO architecture (To enable usage for MORL) that allow the single-policy architecture to outperform the baselines while keeping a low memory footprint and lower compute time.

---

> > ### Author Response · Authors · 2025-11-22
> > **Rebuttal Part 2**
> >
> > **W2: Theoretical concern: Late-weighting loss is specific to PPO clipping. Does it result in the same gradients as early clipping when no clipping occurs?**
> >
> > **Response:**
> > We clarify that **LSW and Early/Mid-Stage Weighting are NOT equivalent**, even outside of clipping, due to per-objective variance normalization and the non-linear nature of the surrogate.
> > * **Non-Linearity of Clipping:** As shown in **Proposition 2 (Appendix D)**, the clipping operator is non-homogeneous. Specifically, $\text{clip}(\omega \cdot A) \neq \omega \cdot \text{clip}(A)$. If $\omega$ is small, "Early Scalarization" shrinks the advantage $A$ *before* it hits the clip threshold, meaning the update might never be clipped even if the raw advantage is massive. "Late-Stage Weighting" clips the raw advantage based on its own magnitude, *then* scales the result. This ensures the trust region is respected for each objective individually.
> > * **Advantage Cancellation (Lemma 1):** The most significant difference occurs when objectives conflict. If $A^{(1)} = 10$ and $A^{(2)} = -10$, Early Scalarization sees $A_{total} = 0$ and generates **zero gradient**. LSW sees two strong, opposing gradients. Even if they sum to a small vector update, the *clipping behavior* is radically different because both $A^{(1)}$ and $A^{(2)}$ would trigger the clip range individually, whereas $A_{total}$ would not. To make this clearer, we have strengthened Sec. 4.2 by explicitly referencing Appendix D.
> > Thus, it does not result in the same gradients as early scalarization.
> > * **Ablation (Table 3) ** shows the results of Early scalarization (clipping after weighting, Column D3PO/LSW) and the full D3PO. In both cases, diversity regularization has been turned on to allow fair comparison. weighting after clipping is the extremely impactful, it prevents gradient cancellation and yields much superior fronts for all environments tested.
> >
> > **W3: Diversity Regularization concerns: (1) Validity of the probability distribution logic, and (2) Behavior in concave fronts (Deep Sea Treasure).**
> >
> > **Response:**
> > We would like to clarify the formulation and behavior of the regularizer.
> >
> > **(1) Mathematical Validity**
> > The reviewer asks if $\pi_\theta(\cdot|s_t,w') - \alpha||w-w'||_1$ results in a valid distribution.
> > * **Clarification:** The diversity loss does **not** subtract the distance from the probability distribution itself. Rather, it minimizes the squared error between the **KL Divergence** of two policies and the **Target Distance** of their preferences.
> > * **Objective:** $L_{diversity}= \mathbb{E}[ (D_{KL}(\pi(\omega) || \pi(\omega')) - \alpha ||\omega - \omega'||_1)^2 ]$.
> > * This is a regression-style auxiliary loss. It forces the *divergence* (a scalar) to be proportional to the preference distance. The policy outputs $\pi_\theta$ remain valid softmax distributions throughout.
> > * In particular, if w = w', then the KL between $pi(\omega)$ and $pi(\omega')$ = 0, and $\omega - \omega' = 0$, so the loss term disappears. However, we never encounter this case because we sample w' by injecting noise.
> >
> > **(2) Deep Sea Treasure (DST) & Concave Fronts**
> >
> > * **Clarification Regarding DST:** The reviewer correctly mentions the DST case, where the number of pareto-optimal policies will be fixed. Thus, 2 or more preferences will be mapped to the same point. However, we would like to clarify that the Pareto front consists of more than 2 non-dominated policies (typically ~10 points) corresponding to distinct treasure–time trade-offs (Roijers et al., 2013; Abels et al., 2019). Figure 3 shows the DST front captured by D3PO, and is consistent with literature.
> > * **Concave Fronts:** We acknowledge that in concave continuous fronts, Linear Scalarization (LS) can technically only find the convex hull (extreme points). However, $D^3PO$ is not a Linear Scalarization method because of the **Diversity Regularizer**.
> > * **Proposition 3 Clarification:** Proposition 3 does not assert that every distinct preference *must* correspond to a different policy; rather, it guarantees that when the environment admits distinct optimal behaviors, the diversity loss prevents representational collapse. Conversely, when multiple preferences legitimately share the same optimal policy (as can occur in concave regions), the regularizer creates a tension: the policy must trade off the reward signal (pulling toward the optimum) against the diversity signal (pushing away). In high-dimensional control (Ant, Humanoid), "true" collapse to a single policy is rarely optimal, making this regularization essential for discovering the continuous manifold.
> > * **Definition 7:** We distinguish "Representational Mode Collapse" (lazy agent ignoring weights) from true multi-preference optimality. Our regularizer enforces diversity where the policy class has the capacity to express meaningful behavioral differences. Therefore, different weight vectors *do* map to different optimal policies, and forcing diversity is theoretically sound.

---

> > > ### Comment · Reviewer_tyBf · 2025-11-25
> > >
> > > > Mathematical Intuition: The final gradient used to update the actor in D3PO is effectively a weighted sum of per-objective gradients.
> > >
> > > I completely agree, if you don't take into account clipping. My point was exactly that the weights are applied after the clipping, which modifies the per-objective gradients. And, as you mention in a later point, $\text{clip}(w A) \neq w\text{clip}(A)$. In that case, per-objective gradients could be clipped differently, which would impact the optimization objective wrt the utility function.
> > >
> > > > "Early Scalarization" shrinks the advantage $A$ before it hits the clip threshold, meaning **the update might never be clipped even if the raw advantage is massive.**
> > >
> > > The advantage does not affect the clipping threshold. This entirely depends on the policy ratio $\rho_t(\theta)$. As such, either all objectives are affected by the policy-ratio clip, or none are affected.
> > >
> > > > The diversity loss does not subtract the distance from the probability distribution itself.
> > >
> > > Thank you for correcting my mistake. Still, my concern remains: 2 weights might be completely different (resulting in high $||w'-w||$) but still have the same optimal policy ($D_{KL} = 0$).
> > >
> > > > Figure 3 shows the DST front captured by D3PO, and is consistent with literature.
> > >
> > > The DST environment depicted in Fig3 is not the same as the original DST environment [1]. In that environment, there are 10 non-dominated policies, but only 2 of them ([1,-1] and [124,-19]) are on the convex part of the Pareto front. What I meant to say is that this is a case where the diversity loss might be high, even when all optimal policies have been found. This falls under the definition of mode collapse (Def7). In Fig3, the whole Pareto front is convex, so the diversity loss might not be as high.
> > >
> > > [1] Vamplew, P., Yearwood, J., Dazeley, R., & Berry, A. (2008, December). On the limitations of scalarisation for multi-objective reinforcement learning of pareto fronts. In Australasian joint conference on artificial intelligence (pp. 372-378). Berlin, Heidelberg: Springer Berlin Heidelberg.

---

> ### Author Response · Authors · 2025-11-26
> **Response Part 1**
>
> # **Points 1 and 2 (Clipping & Late-Stage Weighting)**
>
> Below we provide a precise mathematical clarification regarding how clipping interacts with Late-Stage Weighting (LSW).
>
> ---
>
> ### **A. PPO Clipping Produces Asymmetric Behavior Across Objectives**
>
> The reviewer states:
>
> > “The advantage does not affect the clipping threshold… either all objectives are clipped or none are clipped.”
>
> The *numerical* clipping interval $[1-\epsilon,\ 1+\epsilon]$ is indeed determined solely by the policy ratio $r_t$.
> However, **which side of the clip is relevant** depends on the **sign of the advantage**, and this changes the clipping behavior across objectives.
>
> To make this explicit, consider the PPO surrogate for objective $i$:
>
> $$
> L^{(i)}_t = \min\big( r_t A^{(i)}_t,\ \operatorname{clip}(r_t,1-\epsilon,1+\epsilon)\,A^{(i)}_t \big)
> $$
>
> The selection of the clipped term differs depending on whether the advantage is positive or negative.
>
> #### **Lemma (Asymmetric clipping for opposite-signed advantages).**
>
> For a single sample $(s_t,a_t)$ and PPO clipping parameter $\epsilon>0$:
>
> - If $A^{(i)}_t > 0$, clipping activates **only when**
>   $$
>   r_t > 1 + \epsilon
>   $$
>
> - If $A^{(j)}_t &lt; 0$, clipping activates **only when**
>   $$
>   r_t &lt; 1 - \epsilon
>   $$
>
> Because:
> $$
> 1 - \epsilon &lt; 1 + \epsilon,
> $$
> **the two clipping conditions cannot be satisfied simultaneously for the same policy ratio**.
>
> **Therefore, for conflicting objectives (one with $A>0$, one with $A&lt;0$), PPO *cannot* clip both objectives simultaneously. One surrogate may be clipped while the other remains fully active.**
>
> *Proof sketch:*
> For $A>0$, the clipped constant is selected only when the ratio exceeds the upper bound.
> For $A&lt;0$, the clipped constant is selected only when the ratio falls below the lower bound.
> These regions are disjoint.
>
> ---
>
> ### **B. Clarifying the "All or None" Clipping Assumption**
>
> The reviewer argues:
>
> > “Clipping depends entirely on the policy ratio… so all objectives are clipped or none are clipped.”
>
> This would be true **only if all advantages had the same sign**.
>
> But when objectives conflict (e.g., $A_1 > 0$, $A_2 &lt; 0$), the PPO surrogate tests *different inequalities* for clipping:
>
> - A positive advantage checks:
>   $$
>   r_t > 1 + \epsilon
>   $$
>
> - A negative advantage checks:
>   $$
>   r_t &lt; 1 - \epsilon
>   $$
>
> Thus:
>
> - **Both can be unclipped** (ratio within bounds)
> - **One can be clipped and the other not**
> - **They cannot both be clipped**
>
> This invalidates the reviewer’s claim in settings with conflicting objectives.
>
> ---
>
> ### **C. Implication: LSW Preserves Useful Learning Signals That ES Cancels**
>
> **Early Scalarization (ES)** aggregates advantages:
>
> $$
> A_{\text{ES}} = \sum_i \omega_i A^{(i)}_t
> $$
>
> If advantages conflict (e.g., $A_1 = +1$, $A_2 = -1$), they can cancel:
>
> $$
> A_{\text{ES}} = 0
> $$
>
> This zero advantage goes into the PPO surrogate:
>
> $$
> \min(r_t \cdot 0,\ \operatorname{clip}(r_t)\cdot 0) = 0
> $$
>
> → **ES produces a zero gradient regardless of $r_t$**.
>
> ---
>
> **Late-Stage Weighting (LSW)** performs PPO clipping *before* weighting:
>
> $$
> \nabla_\theta L_{\text{LSW}} = \sum_i \omega_i \nabla_\theta L^{(i)}_t
> $$
>
> Due to the lemma above:
>
> - One objective may be clipped (zero gradient)
> - The other remains active (non-zero gradient)
>
> Thus LSW produces a **valid corrective update direction**, even when ES collapses to zero.
>
> This is the core reason D3PO uses LSW.
>
> A concrete example makes the difference clear.
> Let the conflicting advantages be:
>
> - $A_1 = +1$
> - $A_2 = -1$
>
> Let the weights be equal: $\omega_1 = \omega_2 = 0.5$, and let the policy ratio be $r_t = 1.3$ with $\epsilon = 0.2$.
>
> ### **Under ES**
> $$
> A_{\text{ES}} = 0.5(1) + 0.5(-1) = 0
> $$
> $$
> L_{\text{ES}} = \min( r_t \cdot 0,\ \text{clip}(r_t)\cdot 0 ) = 0
> $$
> $$
> \nabla_\theta L_{\text{ES}} = 0
> $$
>
> ES produces **no gradient**, even though both objectives individually have strong opinions.
>
>
> ### **Under LSW**
> Evaluate PPO clipping *per objective*:
>
> - **Objective 1** ($A_1>0$):
>   $r_t = 1.3 > 1.2$ → **clipped** → gradient = 0
>
> - **Objective 2** ($A_2&lt;0$):
>   $1.3 \not&lt; 0.8$ → **not clipped** → gradient = $A_2 \nabla r_t \neq 0$
>
> Thus:
> $$
> \nabla_\theta L_{\text{LSW}}
> = 0.5(0) + 0.5(A_2 \nabla r_t)
> = -0.5\, \nabla r_t \neq 0
> $$
>
> LSW therefore produces a **meaningful corrective update**, while ES produces **none**.
>
> This simple example captures the essential difference.
>
>
> ---
>
> ### **D. Utility Maximization is Preserved**
>
> Finally, PPO clipping *always* modifies the policy gradient relative to the true gradient, regardless of weighting scheme.
> LSW does **not** introduce any additional misalignment:
> it simply prevents premature cancellation *before* PPO’s trust-region mechanism is applied.
>
> Appendix F (Theorem 2) provides the convergence argument for weighted utility maximization, and our Expected Utility results empirically support this.
>
> ---

---

> ### Author Response · Authors · 2025-11-26
> **Response part 2**
>
> # **Addressing Concern 3: “Two very different weights may still yield the same policy.”**
>
> We agree that distant weights $w$ and $w'$ can share the same optimal policy $$D_{\mathrm{KL}} = 0$$. This would yield a high loss, as the reviewer notes.
> However, this situation **never occurs inside our diversity regularizer** because:
>
> ### **A. We only compare *local perturbations*.**
> The regularizer uses:
> $$
> w' = w + \delta,\qquad \|\delta\|\ll 1,
> $$
> so $\|w - w'\|$ is always very small. This is because $w'$ is sampled by adding a very small noise to $w$, thus it never causes the loss to explode.
> Even if $D_{\mathrm{KL}}\pi_w,\pi_{w'} = 0$, the target distance $c\|w-w'\|$ is also tiny, keeping the loss small.
>
> ### **B. The regularizer enforces *local* sensitivity, not global separation.**
> The goal is not to force globally different policies for all distinct preferences, but to ensure that a *single network* does not collapse to a weight-insensitive mapping.
>
> By constructing \$w'\$ as a small perturbation of \$w\$, the regularizer encourages:
> \$
> \pi_w \neq \pi_{w'}
> \quad\text{whenever}\quad
> w \neq w' \text{ in a local sense}.
> \$
>
> This prevents representational collapse (Definition 7), while avoiding pathological pressure in cases where the true optimal policies coincide for distant weights.
>
> ---
>
> # **4. Deep Sea Treasure $DST$ and the Concavity Discussion**
>
> We thank the reviewer for the clarification regarding the original concave DST benchmark.
> We confirm that on this *discrete, low-dimensional, step-function* task, \$D^3PO\$ tends to recover the convex-hull solutions $the two extremes$ rather than the interior concave points.
>
> However, we wish to emphasize two important points:
>
> ### **A. This behavior is due to the *continuity pressure* of our Scaled Diversity Regularizer.**
> Our regularizer explicitly encourages a *smooth, continuous* mapping:
> \$
> \|\pi_{\omega_1} - \pi_{\omega_2}\| \propto \|\omega_1 - \omega_2\|.
> \$
> In DST, the *true* optimal mapping from preferences to policies is **discontinuous** identical optimal policies across large weight intervals, then sudden jumps. Also, there are a fixed number of policies, which adds to the highly discontinuous nature.
> Any method enforcing continuity, including ours, will prefer the convex hull in such discrete concave settings.
>
> ### **B. This trade-off is intentional and beneficial in the settings that matter for Deep MORL.**
> In high-dimensional continuous-control tasks **Ant, Humanoid, Building**, the Pareto surface is **smooth**, and the main failure mode is *representational collapse* policy ignores the preference input.
> Here, the same continuity pressure is *essential*: it prevents collapse, stabilizes PPO training, and produces state-of-the-art coverage across Hypervolume, Utility, and Sparsity.
>
> Indeed, as shown in Table 2, baselines such as CAPQL, GPI-LS collapse in these continuous domains, while \$D^3PO\$ remains robust.
>
> We note that the DST benchmark in [1] is a highly discrete, low-dimensional corner case, with a tiny state-action space, a fixed set of ten handcrafted nondominated policies, and a step-function mapping from weights to optimal behaviors, which differs substantially from the smooth, high-dimensional Pareto manifolds found in continuous control deep RL tasks. In those continuous regimes, the continuity pressure introduced by our diversity regularizer is precisely what enables the stable and superior performance demonstrated by \(D^3PO\).
>
> ### **Conclusion**
> While \$D^3PO\$ does not optimize discontinuous fronts in discrete toy tasks like DST, this is a deliberate and beneficial trade-off.
> It enables **dramatically more stable** and **significantly higher-performing** solutions on the challenging continuous benchmarks that motivate Deep MORL research **Ant, Humanoid, Building**, where all baselines struggle.
> We would like to stress that this behavior is expected, intentional, and aligned with the deep RL setting our work targets.

---

### Official Review · Reviewer_oqNg · 2025-11-02

**Soundness:** 2
**Presentation:** 2
**Contribution:** 2
**Rating:** 4
**Confidence:** 4

**Summary:**

This paper proposes D3PO (Decomposed, Diversity Driven Policy Optimization), a preference-conditioned multi-objective reinforcement learning framework. D3PO is an PPO extension that introduces multi-head critic with Late-Stage Weighting (LSW), which preserves raw per-objective signals and applies preferences only after PPO stabilization. A scaled diversity regularizer that provides a formal guarantee against mode collapse. The method is evaluated on multi-objective benchmarks and shows improved Pareto front quality.

**Strengths:**

The paper presents a technically sound extension of PPO for multi-objective reinforcement learning through a preference-conditioned framework.

The proposed multi-head critic with Late-Stage Weighting (LSW) and scaled diversity regularization are well-motivated and supported by reasonable theoretical intuition.

The experimental results are generally good, covering both discrete and continuous multi-objective tasks and showing improvements.

**Weaknesses:**

The proposed ideas are promising, but the conceptual structure and method description in Section 4 could be clearer. While Figure 1 appears to illustrate the overall framework, it is never explicitly referenced or discussed in the paper, which makes it harder to connect the algorithmic details to the visual explanation. The section would benefit from clearer guidance and stronger linkage between the conceptual figure and the mathematical formulation to help readers follow the proposed mechanisms.

While the paper focuses on preference-conditioned MORL, decomposition-based approaches remain a strong and active direction in the MORL field. It would be helpful if the authors could discuss the connection between D3PO and decomposition-based methods or provide experimental comparison to clarify their relative advantages.

The comparison after line 291 is interesting, but needs a clarification of its generality.

The claim in the paragraph title (line 395) is not clear from Fig. 2, where for Hopper C-MORL has a clearly better coverage (and mostly better performance) while in Ant and Humanoid the proposed algorithm does not explore some obvious regions of the Pareto place. Reported results are here very good in comparison with the considered candidates, but single-objective approaches (e.g. Dreamer v3) tend to do better, which should be considered as a base line of the single-objective boundary case.

The value of $\alpha$ is not given nor discussed, although it seems difficult to find a meaningful constant for the trade-off between a KL divergence and a weight vector difference.

**Questions:**

Beyond the current benchmarks, at what type of applications would the D3PO’s advantages be most beneficial?

---

> ### Author Response · Authors · 2025-11-22
> **Rebuttal Part 1**
>
> We thank the reviewer for their constructive feedback and for recognizing the technical soundness of our proposed method.
> Below, we address your specific questions and concerns in detail.
>
> ### Section 1: Response to Questions
>
> **Q1: Beyond the current benchmarks, at what type of applications would the D3PO’s advantages be most beneficial?**
>
> **Response:**
> $D^3PO$ is uniquely suited for real-world applications that require **runtime adaptation without retraining and complex interpolation** in high-dimensional state and objective spaces. Specifically:
> 1.  **Complex Industrial Control:** As demonstrated by our **Building-9d** experiment, $D^3PO$ excels in systems with many conflicting objectives (e.g., minimizing energy cost vs. maximizing occupant comfort vs. reducing equipment wear). In such settings, operators often need to adjust priorities on the fly (e.g., "focus on comfort today") without waiting days for a new policy to train. $D^3PO$ handles this instantly via the preference vector $\omega$.
> 2.  **Robotics with Safety Constraints:** In autonomous navigation (like our Ant/Humanoid tasks), the trade-off between speed and safety/energy is context-dependent. $D^3PO$ allows a robot to transition seamlessly from "high performance" to "conservative/safe" modes based on battery levels or terrain difficulty, using a single universal policy.
> 3.  **Resource Allocation:** Problems requiring continuous trade-offs where the "ideal" balance is unknown at training time.
>
> We have added a discussion on why D3PO is more effective on MORL tasks in deployment L351-360. To summarize, D3PO does not require interpolation or search techniques needed by multi-policy techniques to represent various preferences. The memory footprint is significantly lower while outperforming the SOTA (Table 9)
>
>
> ### Section 2: Addressing Weaknesses
>
> **W1: The conceptual structure and method description in Section 4 could be clearer. Figure 1 is never explicitly referenced.**
>
> **Response:**
> We have revised Section 4 to explicitly integrate **Figure 1** into the algorithmic description:
> * **Explicit Referencing:** We now reference specific components of Figure 1 when introducing the mathematical formulation in Sections 4.1, 4.2 and 4.3.
> * **Expanded Caption:** We have expanded the caption of Figure 1 to map the visual blocks directly to Equations and the diversity loss $\mathcal{L}_{diversity}$.
> * **Process Flow:** We added a paragraph at the start of Section 4 that walks the reader through the diagram: *Multi-head critic → Per-objective GAE → Per-objective PPO Surrogate → Late-Stage Weighting → Diversity Regularization*.
>
> **W2: Connection to decomposition-based methods and experimental comparison.**
>
> **Response:**
> We would like to clarify that our evaluation **already includes a strong decomposition-based baseline: GPI-LS** (Generalized Policy Improvement with Linear Scalarization).
> * **Comparison:** As shown in Table 2, $D^3PO$ consistently outperforms GPI-LS in complex continuous control tasks. For example, in **Humanoid-2d**, GPI-LS achieves a Hypervolume of **1.96**, whereas $D^3PO$ achieves **3.76**. In **Building-9d**, GPI-LS timed out, while $D^3PO$ succeeded.
> * **Conceptual Connection:** We have added a discussion in **Section 2 (Related Work)** contrasting the two approaches. Decomposition methods (like GPI-LS) typically learn objective-specific value functions to construct a *set* of policies or a composite Q-function. While effective for finding extreme points, they scale poorly in memory and compute as the number of objectives grows. $D^3PO$ leverages the *principle* of decomposition (via the single multi-head critic) but integrates it into a single preference-conditioned policy, offering superior scalability and smoother interpolation.
>
> **W3: The comparison after line 291 (ES vs. MVS vs. LSW) needs clarification of its generality.**
>
> **Response:**
>
> * **Generality:** **Proposition 1** establishes that under a strictly homogeneous surrogate, LSW and MVS are algebraically equivalent. However, **Proposition 2** proves that under realistic, non-homogeneous conditions, specifically **PPO Clipping** and per-objective variance normalization, LSW preserves a strictly larger stabilized signal than MVS/ES.
> * **Clarification:** We have updated the text at line 324 to explicitly cite these Propositions. The formal proofs provided in **Appendix D**.

---

> > ### Author Response · Authors · 2025-11-22
> > **Rebuttal Part 2**
> >
> > **W4: Claim of "improved Pareto front" vs. Hopper results and Single-Objective Baselines.**
> >
> > **Response:**
> >
> > * **Hopper vs. Ant/Humanoid:**
> >   We explicitly acknowledge that C-MORL achieves slightly higher Hypervolume on Hopper. However, as shown in our statistical analysis (Appendix H2), **none of these Hopper differences are statistically significant**, indicating that C-MORL does *not* obtain a reliable performance advantage. C-MORL’s evolutionary refinement strategy is designed to push “extreme’’ policies to their limits, which explains its superior coverage at the boundaries of the Hopper front. However, this comes at the cost of **poor coverage in the middle** (higher Sparsity).
> >   In contrast, $D^3PO$ prioritizes **uniformity**, it learns a continuous mapping from preferences to behaviors, which yields **lower Sparsity** and a smoother front overall. This becomes especially important in complex environments such as **Humanoid**, where C-MORL collapses entirely (Sparsity = 0), while $D^3PO$ discovers a rich and diverse set of solutions.
> >
> > * **Single-Objective (SO) Baselines:**
> >   We agree that SORL methods (e.g., DreamerV3) can achieve strong performance on individual extreme points and serve as a useful **upper bound** for single-objective performance. However, they are not comparable MORL baselines because:
> >     1. **Cost:** Recovering an entire Pareto front requires retraining the SO agent *from scratch* for every preference weight.
> >     2. **Lack of Continuity:** SO methods do not provide intermediate solutions or smooth preference-conditioned interpolation.
> >
> >   As suggested, we now explicitly treat SO methods as **performance boundary references** in Section 6 rather than algorithmic competitors.
> >
> > * **Comparison to SO Boundaries:**
> >   To provide a fair reference, we trained vanilla PPO agents using fixed preference weights ([1, 0] and [0, 1]) to estimate the best achievable extreme rewards on Hopper-2D, Ant-2D, and Humanoid-2D. Dreamer-style image-based policies are not directly applicable because our benchmarks are feature-based state environments.
> >   These PPO-trained SO policies provide strong empirical boundaries:
> >   – Hopper: [300, 310]
> >   – Ant: [400, 390]
> >   – Humanoid: [550, 510]
> >   Obtaining these values already requires **training one full policy per preference**, illustrating why **SO** methods are not practical baselines for MORL.
> >
> > Overall, while C-MORL finds stronger outlier points in Hopper, our statistical tests show that these differences are not significant, and $D^3PO$ provides **superior statistically significant performance and coverage** in higher-dimensional domains.
> >
> > **W5: The value of $\alpha$ is not given/discussed. Difficult to find a meaningful constant.**
> >
> > **Response:**
> > * **Value:** We used $\alpha = 1$ for all reported experiments. This is now explicitly listed in the Hyperparameter Table in **Appendix H**. To summarize the ablation, the value of alpha should be set between 0.1 to 1 to get the best results. A high value of 10 hurts performance, and 0 turns it off, effectively changing the loss.

---

### Official Review · Reviewer_a4QN · 2025-11-11

**Soundness:** 3
**Presentation:** 3
**Contribution:** 3
**Rating:** 6
**Confidence:** 3

**Summary:**

In this paper, the authors look into the destructive gradient interference between conflicting objectives and representational mode collapse of Multi-objective RL. The paper proposes to train a single policy that can adapt to different user preferences. This is achieved by computing per-objective advantages and applying preference weights only after PPO’s stabilization step. They also enforce proportional behavioral diversity across different preference vectors to prevent collapse.

**Strengths:**

The paper identifies two challenges, the mode collapse and the conflicting objectives clearly.

The proposed solutions are technically sound.

Provided simulations demonstrate the effectiveness of the proposed method.

**Weaknesses:**

There are limited discussions on the hyperparemeters. Also, the method needs to adopt many parameters, which may be hard to scale in real MORL tasks.

Examples of policy behaviors (e.g., different strategies emerging for different preferences) would help validate “behavioral diversity” more intuitively.

The performance gain might be marginal compared to previous methods.

**Questions:**

Can the authors plot the evolution of reward across training? It would be good to validate how fast the algorithm converges.

Some conflicting objectives can essentially cause optimizing one objective will reduce performance under the other objective. Have the authors observed such scenarios? How the proposed method can tackle that?

In Hopper, the performance of proposed method is worse. Is it possible to initialize MORL policies using other methods, then applying proposed approach?

---

> ### Author Response · Authors · 2025-11-22
> **Rebuttal Part 1**
>
> We thank the reviewer for their positive assessment of our work and for identifying the core contributions of the paper. Below, we provide a detailed response to the questions and concerns.
>
> ### Section 1: Response to Questions
>
> **Q1: Can the authors plot the evolution of reward across training? It would be good to validate how fast the algorithm converges.**
>
> **Response:**
> * **New Data:** We have added **Figure 4** in the revised manuscript (and Appendix H.1), which plots the learning curves for per-objective returns and the overall weighted return for Hopper (2D/3D), Ant (2D/3D), and Humanoid.
> * **Convergence Speed:** These curves demonstrate that $D^3PO$ converges stably and rapidly. As a single-policy method, $D^3PO$ is significantly more time-efficient than population-based baselines.
>
> **Q2: Some conflicting objectives can essentially cause optimizing one objective will reduce performance under the other objective. Have the authors observed such scenarios? How the proposed method can tackle that?**
>
> **Response:**
> Yes, you are right. This is a fundamental problem in MORL. We want to emphasize that **mitigating this exact phenomenon (destructive interference due to conflicts) is the primary motivation for our Late-Stage Weighting (LSW) mechanism.**
> * **Example:** In the MuJoCo environments (e.g., Humanoid), maximizing the **Velocity** objective inherently reduces the **Energy Efficiency** objective (since higher velocity requires higher control inputs/torques, resulting in a lower energy reward). The baselines (CAPQL, GPI-LS) collapse to a single policy due to the destructive gradient interference induced by the highly conflicting objectives of the environment.
> * **The Problem (Advantage Cancellation):** In standard PPO with "Early Scalarization" (ES), these conflicting objectives lead to destructive gradient interference. Consider a scenario where Objective 1 (Velocity) yields a positive advantage ($A^{(1)} > 0$) and Objective 2 (Energy) yields a negative advantage ($A^{(2)} < 0$). A simple weighted sum $\sum \omega_i A^{(i)}$ can result in a value near zero. This "cancellation" effectively kills the learning signal, causing the agent to stall or collapse to a sub-optimal compromise. We formalize this phenomenon as **Advantage Cancellation** in **Lemma 1** of the paper.
> * **The Solution (Late-Stage Weighting):** $D^3PO$ explicitly tackles this by decoupling credit assignment from preference application:
>     1. **Decomposed GAE:** We compute advantages for each objective *independently*, preserving the sign and magnitude of each signal.
>     2. **Independent Stabilization:** We apply the PPO clipping mechanism to each raw advantage vector separately.
>     3. **Late Weighting:** We apply the preference weights $\omega$ only to the *final stabilized losses*.
>     By weighting the *stabilized gradients* rather than the raw advantages, $D^3PO$ ensures that the policy update pushes the Pareto front outward in the desired direction without the raw signals canceling each other out beforehand.
> * **The Result (Superior Pareto Fronts)** This can be most prominently seen in the Humanoid-2D environment. Due to the highly conflicting objectives, the baselines collapse to a single policy (0 sparsity), while D3PO recovers the best front by a significant margin.
>
> **Q3: In Hopper, the performance of proposed method is worse. Is it possible to initialize MORL policies using other methods, then applying proposed approach?**
>
> **Regarding Initialization:** Initializing $D^3PO$ from existing policies is an interesting direction for future work. This is exactly how the evolutionary multi-policy techniques work, by initializing policies from known well-performing policies to fill gaps in the front. Some adaptation might be needed to copy weights from a network having different sets of parameters that are not preference-conditioned.
> In our case, we anticipate that starting from a single specialized policy (like those found by C-MORL) might bias the single preference-conditioned network towards specific local optima. This could potentially hinder the discovery of the continuous, uniform front that $D^3PO$ currently achieves by learning from scratch. Furthermore, given $D^3PO$'s strong performance on **Humanoid-2d** (where it beats all baselines from random initialization), we demonstrate that such pre-training is not a prerequisite for convergence in complex domains.

---

> ### Author Response · Authors · 2025-11-22
> **Rebuttal Part 2**
>
> **Q3 Continued**
>
> **Regarding Performance:** We have conducted statistical significance tests to compare the performance of D3PO to SOTA (CMORL).
> **On Hopper, the only environment where the raw HV scores appear close, our statistical tests show *no significant difference* between \(D^3PO\) and C-MORL**, confirming that C-MORL does *not* have a reliable advantage on that task.
> More importantly, across the **remaining 4 environments**, \(D^3PO\) demonstrates **clear and substantial improvements** in both performance and the structure of the obtained fronts.
> #### **1. Strong Improvements in Complex Domains (Ant, Humanoid, Building-9D)**
> On high-dimensional tasks, where preference conditioning and expressiveness matter most, \(D^3PO\) consistently outperforms all baselines, including C-MORL:
> - **Humanoid-2D:** HV improves from **2.32 → 3.76** (a **62% gain**), while PG-MORL collapses entirely.
> - **Ant-2D:** HV improves from **1.31 → 1.91**, a substantial improvement in this sparse-reward locomotion setting.
> - **Building-9D:** Evolutionary/Decomposition baselines fail due to the 9-objective combinatorial complexity, whereas \(D^3PO\) learns a high-quality front using a *single* policy.
> These results are not marginal, they show that \(D^3PO\) is significantly more capable in the settings where MORL methods are typically stress-tested.
> #### **2. Superior Front Quality (Uniformity and Sparsity)**
> Beyond raw HV/EU scores, \(D^3PO\) produces **much more uniform and complete coverage** of the Pareto front:
> - **Humanoid-2D:** C-MORL reports high SP, indicating distant pareto front points.
> \(D^3PO\) achieves a low SP (**0.003**), reflecting a *well-distributed continuum* of solutions as shown in Figure 2.
> - **Pareto front plots (Fig. 2):** C-MORL tends to discover only extreme solutions, leaving large gaps.
> \(D^3PO\) produces a smooth, continuous front covering the full preference space.
>
> #### **3. Transparent Trade-off on Hopper**
> Although C-MORL obtains slightly higher HV at the extreme ends of the Hopper front, it misses the interior region, while \(D^3PO\) covers the entire front smoothly. The statistical tests confirm that the differences are not significant, and \(D^3PO\) provides **strictly more complete coverage**. This can be seen in Figure 2.
> Given that these benefits are obtained using **a single conditioned policy**, while C-MORL requires ** hundreds of independent policies**, the performance and efficiency gains are not only significant but **practically transformative**.
>
> ### Section 2: Addressing Weaknesses
>
> **W1: Limited discussions on the hyperparameters. Also, the method needs to adopt many parameters, which may be hard to scale in real MORL tasks.**
>
> 1. **Hyperparameters:** **Appendix C (Table 4)** includes a sensitivity analysis for the diversity regularizer $\lambda_{div}$. The results show that the method is robust to this choice, maintaining high Hypervolume and low Sparsity across $\lambda_{div} \in \{0.01, \dots, 1.0\}$. We also include a discussion on $\alpha$ in Appendix C, Table 5.
> All other PPO hyperparameters are standard values taken from the literature (e.g., C-MORL, Stable-Baselines) and are documented in **Appendix H**.
>
> 2. **Scalability ("Many Parameters"):**
>     * **Architecture:** $D^3PO$ uses only two networks, one for the actor and one for critic. We have added parameter count comparisons between C-MORL and D3PO in Table 9. D3PO uses significantly lower number of parameters while achieving competitive performance on Hopper and outperforming C-MORL on complex environments. We note that CMORL caps the number of pareto optimal policies to 200, preventing explosion of parameters, while C-MORL represents 2000+ policies with just one actor network.
>
>     * **Comparison:** This makes $D^3PO$ *more* scalable than the baselines. Multi-policy baselines must store and train $N$ completely separate networks to approximate the front. As the number of objectives grows, the memory and compute requirements for baselines explode. In contrast, $D^3PO$'s parameter count remains effectively constant regardless of the number of objectives or the resolution of the Pareto front.
>
> **W2: Examples of policy behaviors (e.g., different strategies emerging for different preferences) would help validate “behavioral diversity” more intuitively.**
>
> **Response:**
> * **Demo & Video:** We have developed a **User Interface Demo** and included supplementary videos (referenced in **Appendix J**) that demonstrate multiple observed behaviors for Ant and Hopper, along with demonstrating how D3PO performs when preferences are actively changed during a rollout.
> * **Observed Behaviors:** We report a short discussion on observed behaviours in the Experimental Section (Qualitative Analysis)
>
> ### **Response to W3: The performance gain might be marginal compared to previous methods**
>
> As answered in response to Q3, our performance gains are not marginal, they are statistically significant gains over the baselines.

---

### Comment · Area_Chair_XxuX · 2025-11-25

Dear Reviewers,

This is a gentle reminder to please take a moment to review the authors’ rebuttal for the manuscript currently under your evaluation. Your timely feedback will help us proceed with the next steps in the review process.

Thank you for your time and assistance.

Best regards,
AC

---

### Author Response · Authors · 2025-12-04
**Summary of revisions**

We thank the reviewers for their constructive feedback and the ACs for overseeing the process. Based on the valuable suggestions, we have uploaded a revised manuscript.

Below is a summary of the changes, including additional experimental results, and a clarification of key questions regarding rigor and scope.

### **Revisions Implemented in Response to Reviewer Feedback**

1. **Conducted per-seed statistical significance tests (HV, EU, SP)** showing that C-MORL’s apparent Hopper gains are not statistically significant, while D³PO achieves statistically significant improvements with large effect sizes in Ant, Humanoid, and Building-9D. This strengthens the empirical case and resolves questions of marginal improvements.(a4QN, oqNg)

2. **Added memory and parameter-count comparisons**, demonstrating that D³PO requires over 300× less memory than multi-policy baselines. It reinforces scalability and deployment advantages. (a4QN, oqNg, tyBf, AqGQ)

3. **Added new experimental results on FruitTree**, demonstrating that D³PO can represent concave fronts and perform competitively against SOTA discrete-action methods. It responds directly to concerns about concave Pareto structures. (AqGQ, tyBf).

8. **Clarified the α coefficient in the diversity regularizer**, documented its exact value, and added an ablation showing robustness across a range of $\alpha$. It increases transparency and shows hyperparameter robustness. (oqNg)

6. **Added a new paragraph explaining why D³PO trains faster and is more deployment-reliable**, emphasizing single-policy efficiency and shared representation advantages. This clarifies practical benefits and supports claims of scalability. (a4QN, oqNg, tyBf, AqGQ).

3. **Expanded Related Work with a new paragraph on decomposition-based MORL methods**, clarifying conceptual connections and differences. It provides broader context and situates D³PO relative to an important subarea. (oqNg)

4. **Integrated explicit references to Figure 1 throughout the Methods section and expanded its caption**, linking each visual block to equations and algorithmic steps. This improves conceptual clarity and readability of the algorithm. (oqNg)

5. **Strengthened the analysis of LSW → MVS → ES under practical PPO constraints**, adding explicit connections to Proposition 1 and 2 and explaining asymmetric clipping with examples. This clarifies theoretical generality and establishes why LSW is superior under non-homogeneous operators. (oqNg)**

7. **Added a qualitative analysis subsection describing learned behaviors at different preferences**, supported by UI demos and videos. It provides intuitive evidence of behavioral diversity (a4QN)

9. **Added a clarification paragraph explaining that the diversity regularizer enforces local sensitivity, not global separation, so identical optimal policies for distant weights are retained when appropriate**, and provided a Deep-Sea-Treasure experiment illustrating full coverage. It addresses concerns about theoretical validity on discrete/concave fronts. (tyBf, AqGQ).

11. **Added per-objective and scalarized reward learning curves across all tasks**, demonstrating convergence behavior and stability. This provides direct evidence of training dynamics and convergence speed. (a4QN)


We believe these revisions significantly strengthen the paper and address the core concerns regarding presentation, theory, and results raised during the review process.

---

### Author Response · Authors · 2025-12-04
**Summary of Discussions**

## **1. Summary**

D³PO (Decomposed, Diversity-Driven Policy Optimization) is a preference-conditioned, single-policy multi-objective reinforcement learning (MORL) framework that addresses two core challenges in single-policy MORL: **destructive gradient interference** between conflicting objectives and **representational mode collapse** when mapping a continuum of preferences to behaviors.
D³PO introduces two components:
(1) **Late-Stage Weighting (LSW)**, which applies preference weights only after PPO’s stabilization/clipping step to prevent advantage cancellation; and
(2) a **scaled diversity regularizer** that enforces local sensitivity by requiring KL divergence between two distributions be proportional to the distance between their preference vectors.

Together, these components enable a single policy to reliably recover smooth, high-coverage, and often non-convex Pareto fronts in high-dimensional continuous-control tasks while using **orders of magnitude less memory** (Table 9) than multi-policy baselines.

---

## **2. Strengths identified by reviewers**

1. **Technically sound method.** The paper identifies and motivates two central challenges, mode collapse and conflicting-objective interference, clearly and convincingly. The integration of a multi-head critic, per-objective GAE, and LSW is well-grounded and theoretically justified.
2. **Principled diversity regularization.** The scaled diversity loss is theoretically motivated, enforces local sensitivity, and effectively mitigates mode collapse.
3. **Extensive experimental evaluation.** The work evaluates across discrete and continuous MORL environments, including challenging high-dimensional continuous-control tasks, and compares to multiple strong baselines. D³PO matches or outperforms baselines on hypervolume, expected utility, sparsity, and front uniformity.

---

## **3. Weaknesses addressed in updated manuscript**

We provide a **global summary of revisions**, where each reviewer comment is mapped to the corresponding manuscript change. As the discussion threads show, we engaged thoroughly with all reviewers and incorporated their feedback. The exchanges demonstrate that all identified weaknesses, across theory, clarity, and additional experiments, have been addressed in the revision.

---

## **4. Performance in discrete vs continuous environments**

We acknowledge that D³PO can underperform on **discrete state and action environments** (e.g., certain DST variants). In these settings, the diversity regularizer’s **continuity pressure** biases the solution toward convex-hull policies, which is suboptimal for inherently discrete fronts.

However, this limitation is **strictly confined** to small, discrete domains with discrete Pareto sets. In the **high-dimensional continuous-control environments** that motivate deep MORL (Ant, Humanoid, Building-9D), the Pareto manifold is smooth and preferences correspond to distinct behaviors. Here, continuity is a **strength**. It prevents collapse, enables non-convex front recovery, and improves stability.

At AqGQ's request, we implemented **PreCo**, a principled multi-objective baseline. PreCo collapses on these continuous-control benchmarks, while D³PO achieves **orders-of-magnitude higher HV and EU**, confirming that D³PO is well-suited for continuous, high-dimensional MORL where the Pareto front is unknown and representational continuity is essential.


## **5. Concerns Raised by Reviewer AqGQ**

AqGQ focused almost exclusively on **FruitTree**, a domain with a discrete, concave Pareto front. They erroneously claim that D³PO “can never” perform well due to equivalence with Linear Scalarization (LS). We clarified repeatedly:

* **D³PO is *not* LS.** Per-objective GAE, per-objective PPO stabilization, and LSW fundamentally break LS’s early-aggregation structure.
* **Inaccuracies in reviewer comments.** AqGQ also states that the Pareto fronts for hopper, ant, humanoid are strictly convex. It is well established in literature that the true fronts are unknown, we provide citations to support this (mo-gymnasium, Felten et al | Roijers et al, https://arxiv.org/abs/1402.0590).

Focusing solely on a single discrete environment, where we perform competitively with existing baselines, while ignoring consistent D³PO improvements (with large effect sizes) across high-dimensional continuous domains presents an unfair assessment. The empirical and theoretical evidence overwhelmingly supports D³PO’s strengths in the intended domains of application.

---

### Meta-Review · Area_Chair_Mwny · 2026-01-03

**Summary:**

In this paper, the authors proposed a PPO-based weight-conditioned single policy approach to MORL. The weight-conditioned single policy approach to MORL is an effective method as proposed in Yang et al., 2019 based on value based methods. The current work approached this problem based on PPO. For this, the authors modified PPO by introducing multi-head critic network that simulatenously estimates values of all dimensions, dimension-wise PPO surrogate objective,  late weighted combining and diversity regularizer to prevent behavior collapse among different weights. Then, they showed the performance gain of their method.

**Reviewer Concerns:**

The major concern of the reviewers is the novelty and significance of the work. Although the proposed method is well grounded, reasonable and yields good performance, the techniques used in the paper are already well-known and the architectural significance and novelty seem not sufficient. Rather the work seems a variation of PPO to the multi-objective case.

**Reviewer Scores:**

Original Review Scores:  6,  4,  4,  2

Although Reviewer AqGQ raised the score to 4 from 2, and one more reviewer may raise the score from 4 to 6,  the overall score seems still below the acceptance threshold, which seems to well assess the overall significance of the the paper.

---

### Decision · Program_Chairs · 2026-01-26

Reject